# Structural and Histomorphological Evaluation of the Stifle Joint Capsule in Canine Congenital Patellar Luxation and Cranial Cruciate Ligament Rupture

**DOI:** 10.3390/ani15243545

**Published:** 2025-12-09

**Authors:** Mario Candela Andrade, Petra Peer, Pavel Slunsky, Matias Aguilera-Rojas, Johanna Plendl, Leo Brunnberg

**Affiliations:** 1Department of Medicine, Health and Medical University Potsdam, 14471 Potsdam, Brandenburg, Germany; 2South Tyrolean Health Authority, Veterinary Service, Health District of Brixen, 39042 Brixen, South Tyrol, Italy; 3AniCura Small Animal Clinic Augsburg, 86157 Augsburg, Bavaria, Germany; 4Institute of Veterinary Anatomy, Center for Biomedical Sciences, School of Veterinary Medicine, Freie Universität Berlin, 14195 Berlin, Germany; 5Small Animal Clinic, Department of Veterinary Medicine, Freie Universität Berlin, 14163 Berlin, Germany

**Keywords:** knee, dislocation, tissue, dog, histomorphology, PL, CCLR, stifle joint capsule

## Abstract

Patellar luxation (PL) and cranial cruciate ligament rupture (CCLR) are common causes of lameness in dogs, often linked to inherited joint weakness. This study examined the microscopic structure of the stifle joint capsule in affected dogs to understand how these conditions alter joint tissues. Compared with healthy dogs, those with PL or CCLR had noticeable changes in the joint capsule layers and fewer normal surface folds, especially in long-term cases. These findings suggest that ongoing joint instability is associated with distinct alterations of the joint capsule, particularly affecting the composition and organization of its layers. Recognizing these early tissue changes may help veterinarians improve diagnosis, treatment, and breeding strategies to reduce the risk of joint problems in predisposed dog breeds.

## 1. Introduction

Congenital patellar luxation (PL) and cranial cruciate ligament rupture (CCLR) are among the most prevalent hereditary musculoskeletal disorders in dogs [1,2,3,4]. Patellar luxation, which is medially directed in most cases, is considered a multifactorial condition involving both skeletal malformations and soft tissue abnormalities, such as joint capsule laxity. Similarly, CCLR is influenced by multiple factors, including an excessive tibial plateau angle, being overweight, older age, breed predispositions, and progressive degeneration of the ligament [2,3,4]. While skeletal changes associated with patellar luxation—such as femoral trochlear hypoplasia, tibial tuberosity deviation, and a shallow trochlear sulcus—are well documented, the role of soft tissue structures, particularly the stifle joint capsule, remains underexplored.

Recent discoveries, such as the identification of a novel elastic fiber layer in the human shoulder joint, underscore how much remains unknown about soft tissue microstructures and highlight the need for further research [5]. Studies in human medicine show that connective tissue defects, such as those affecting collagen and elastin, contribute to joint instability [6]. Drawing on these findings, it is hypothesized that connective tissue abnormalities in the stifle joint capsule may exacerbate the laxity observed in congenital patellar luxation in dogs, complementing known skeletal abnormalities [7,8]. However, whether this soft tissue laxity is an integral part of the multifactorial etiology or a consequence of the skeletal malformations and patellar luxation remains unknown.

The aim of this study is to investigate congenital patellar luxation and cranial cruciate ligament rupture (CCLR) through a detailed histomorphological analysis. Specifically, it will characterize the structure of a healthy canine stifle joint capsule, identify structural deviations in the joint capsules of dogs affected by PL and/or CCLR, and examine the organization of tissue layers, blood vessels, cells, and cellular products. Additionally, differences in the distribution patterns of these structures will be explored.

During surgical correction procedures, stifle joint capsule samples from affected dogs will be collected for histological examination. A qualitative analysis, combined with a comparison to other conditions and healthy tissue samples, will provide critical insights into the previously understudied microstructural conditions of the canine stifle joint capsule. This approach will enhance our understanding of the role of soft tissue abnormalities in both congenital patellar luxation and CCLR, highlighting potential shared and distinct pathological mechanisms between these two disorders.

## 2. Materials and Methods

### 2.1. Patients

This study included dogs diagnosed and surgically treated for medially directed patellar luxation (PL) or cranial cruciate ligament rupture (CCLR), both of which required the removal of a small portion of the stifle joint capsule to facilitate a lateral “capsular imbrication” procedure. Additionally, dogs with distal femoral fractures, which also necessitated capsular imbrication, and those euthanized due to unrelated health issues (e.g., foreign body ingestion or age-related illness) were included as controls. The animal owners provided written consent for the removal of a stifle joint capsule sample for this study.

All dogs were categorized into the following groups:

Study groups:
-PL group: Dogs diagnosed and surgically treated for PL.-CCLR group: Dogs diagnosed and surgically treated for CCLR.-Patellar luxation and concomitant cranial cruciate ligament rupture group: (PL + CCLR): Dogs diagnosed and surgically treated for both conditions.
Control group:
-Dogs euthanized for unrelated health issues or surgically treated for distal femoral fractures that required an arthrotomy due to hemarthrosis.


Inclusion criteria included dogs diagnosed with patellar luxation (PL) confirmed through physical examination and graded or dogs diagnosed with cranial cruciate ligament rupture (CCLR) based on physical and radiological assessments. Additional factors such as breed, sex, direction of luxation, lameness duration, and age were also considered. Exclusion criteria included incomplete clinical records, inconsistent grading of patellar luxation, inability to obtain joint capsule samples, or the presence of extra-skeletal systemic diseases that could affect joint status, such as polyarthritis, neoplasia, or systemic infections (e.g., sepsis). The study included dogs of various breeds, with the sample classified according to the American Kennel Club standards [9] and further divided into weight classes in 5 kg increments, starting from 0–5 kg up to >45 kg. The duration of lameness was categorized as follows: peracute (1–2 days), acute (3–14 days), subacute (15–30 days), subchronic (31–90 days), and chronic (>90 days).

Patellar luxation was graded using the Singleton classification [10], which ranges from Grade I to IV. Grade I represents intermittent luxation with the patella returning to its normal position spontaneously. Grade IV indicates permanent luxation with the patella fixed outside the trochlear groove and severe skeletal deformities. Intermediate grades (II and III) reflect increasing severity, including frequent luxation and reduced patellar stability (Appendix A). Studies have shown that different veterinarians may assign varying patellar luxation grades to the same patient [4,11]. Therefore, when different patellar luxation grades were recorded for the same affected stifle joint by multiple veterinarians, cases were classified into intermediate grades based on the clinical documentation in order to reflect the most clinically relevant severity level.

CCLR was diagnosed using a combination of clinical signs, including pain during stifle manipulation, joint effusion, variable crepitus, medial buttress formation, and a positive cranial drawer test, supported by radiographic evaluation according to Kowaleski et al. [10]. For the control group, joint capsule samples were obtained from dogs euthanized for unrelated health issues, such as foreign body ingestion or age-related conditions. Orthopedic disease was ruled out through physical examination, clinical history, and owner-reported history. Owners provided written consent for the use of these samples.

Two animals were euthanized due to unrelated health issues, both with a poor prognosis. One had an inoperable ileus caused by a ruptured intestine following foreign body ingestion and was euthanized intraoperatively. The other suffered from chronic kidney failure and was also euthanized due to the unfavorable prognosis. In both cases, euthanasia was performed at the owners’ request. To ensure minimal distress, euthanasia was carried out following appropriate protocols. All animals were premedicated with midazolam (0.1–0.3 mg/kg IV, Midazolam^®^; B. Braun Melsungen AG, Melsungen, Germany) and induced with propofol (4 mg/kg IV, Narcofol^®^; CP-Pharma, Burgdorf, Germany) via a pre-placed intravenous catheter. Pentobarbital (50–60 mg/kg IV, Narcodorm^®^; CP-Pharma, Burgdorf, Germany) was then administered as a bolus injection, leading to a rapid and painless loss of consciousness, with death confirmed by the absence of vital signs.

### 2.2. Examination Material

Small portions of the stifle joint capsule were excised laterally and longitudinally using a scalpel (Aesculap^®^, Aesculap, Inc., Center Valley, PA, USA) to create 0.5–2 cm^2^ incisions. The excisions were made laterally parapatellar, with attention to the vastus lateralis muscle, which demarcates the proximal end as it extends into the joint capsule. The distal end was marked at the level of the distal end of the patella. This procedure facilitated a “capsular imbrication” [12] as part of the treatment for PL and CCLR cases, involving the tightening of the joint capsule laterally by overlapping or suturing its edges to enhance joint stability (Figure 1). Dogs with CCLR were also treated with a fascial imbrication or Meutstege technique [3,12],

### 2.3. Histological Analysis

All joint capsule samples were prepared at the Institute for Veterinary Anatomy (Center for Biomedical Sciences, School of Veterinary Medicine, Freie Universität Berlin) following the guidelines of Mulisch and Welsch [13]. This standardized procedure included fixation, dehydration, embedding, and sectioning, ensuring optimal preservation for microscopic examination. Samples were stained using four histological techniques and analyzed under light microscopy multiple times (Appendix A).

### 2.4. Histological Overview Staining

Four histological stains were performed: Hematoxylin-Eosin, Combined Weigert-Masson-Goldner, Combined Elastin (Weigert-Volkmann-Strauß), and Fibrin Stains (Ladewig). Each biopsy had at least three slides prepared for each stain, allowing for comparison across samples. These four stains were also based on guidelines from Mulisch and Welsch [13].

#### 2.4.1. Hematoxylin-Eosin Staining

Sections were stained with Mayer’s Hematoxylin, counterstained with eosin, and mounted with coverslips. This method stains nuclei and cytoplasm distinctly, providing a clear overview of tissue structure [13].

#### 2.4.2. Combined Weigert-Masson-Goldner Staining

This method differentiates connective tissue components using multiple dyes, allowing for detailed analysis of the joint capsule structure and pathology. The staining process involves steps such as staining with Weigert’s hematoxylin, Masson’s trichrome components, to ensure clear contrast between tissue elements [13].

#### 2.4.3. Combined Weigert-Volkmann-Strauß Staining

Since the elastic fibers are not distinguishable from the collagen fibers using the combined Weigert-Masson-Goldner stain, the Volkmann-Strauß stain was additionally employed. To simultaneously visualize elastin, muscle, and collagen, particularly for distinguishing vascular walls, Volkmann and Strauß recommend a method based on Heidenhain’s Azan technique. In this method, Resorcin-Gentian violet is used for staining elastin, Azocarmine G for staining nuclei, plasma, and muscles, and Naphthol Green B for staining collagen. Between these staining steps, the samples are treated with 70–95% alcohol and water, and alternated with phosphotungstic acid. The samples are mounted as usual after stepwise dehydration in alcohol [13].

#### 2.4.4. Fibrin Detection According to Ladewig

The Ladewig stain provides a clearly highlighted fibrin stain, which can indicate pathological changes. This staining method can be applied to any fixed material and is highly stable. It produces uniform and precipitate-free specimens with very consistent results. The overall image is very clear due to the refinement of contrasts, facilitating the assessment of pathological tissue changes in the context of inflammation.

#### 2.4.5. Morphometry

From the 78 stifle joint capsule samples, a total of 1404 histological sections were prepared for analysis. To verify the findings, three slides with two sections per staining were produced for the histological examination. The evaluation was carried out using a Zeiss Axioskop 40 microscope (Zeiss, Oberkochen, Germany) with various objectives (1.25×, 5×, 10×, 20×, 40×). The results of the sections were quantified and photographically documented using NIS-Elements AR^®^ software, version 4.5 (Nikon, Chiyoda, Tokyo, Japan; Nikon GmbH, Düsseldorf, Germany). The following section describes the structures to be quantified and the method of their quantification.

### 2.5. Explanations of the Individual Examinations

The stifle joint capsule in the histological section can be divided into a proximal section with a cross-section of the quadriceps femoris muscle, a middle section that is distinctly tapered, and a distal section. The vastus lateralis muscle of the quadriceps femoris extends into the proximal stratum fibrosum. Its muscle fiber bundles, mostly seen in cross-section, stain dark red-violet in the Volkmann-Strauß staining, thus distinguishing them from the surrounding, dark green-stained and strictly parallel-arranged collagen fiber bundles of the joint capsule (Figure 2). With this finding, the proximal section (Figure 2A) could be clearly differentiated from the other two sections (Figure 2B,C).

The histological sections were used for multiple thickness measurements at 90 degrees to the inner surface of the joint capsule in the proximal, middle, and distal sections of the joint capsule samples (Figure 2). Histomorphological criteria were established to distinguish between the three main layers of the joint capsule: the stratum synoviale, stratum subsynoviale, and stratum fibrosum.

The stratum synoviale, the innermost layer, is characterized by a cell-rich superficial layer lining the joint cavity. Adjacent to this is the stratum subsynoviale, featuring well-vascularized connective tissue with fine collagen fiber bundles, venules, arterioles, and occasional fat cells extending into synovial plicae. Elastic fibers are also present in this layer. The outermost layer, the stratum fibrosum, is distinguished by thick collagen fiber bundles and is relatively cell-poor. This layer contains varying amounts of fat cells, particularly in the proximal section associated with the retropatellar fat pad, and houses blood vessels and nerves interspersed within its dense collagen network [14].

These distinct histological features were used to identify and measure the thickness of each layer across the proximal, middle, and distal sections of the joint capsule. This approach allowed for a comprehensive analysis of the structural variations along the length of the stifle joint capsule. The four staining methods (Volkmann-Strauss, Weiger-Masson-Goldner, Ladewig, and Hematoxylin-Eosin) were used to quantify the following parameters: the number of superficial cell layers in the stratum synoviale, the thickness of the capsule, and the thickness of the stratum fibrosum, subsynoviale, and synoviale in the proximal, middle, and distal sections. Additionally, the average length of at least six villi and the surface length of the villi section were measured. Using Ladewig staining, the surface area proportion of fibrin in the joint capsule and the localization of fibrin deposits across the proximal, middle, and distal sections were evaluated.

Regarding the villous surface, the length of the superficial cell layer contour of the stratum synoviale was measured (Figure 3, orange line). A straight line parallel to the inner surface of the joint capsule was then measured (Figure 3, red line). Using these two values, the surface enlargement factor was calculated (villous section surface length/corresponding straight line = surface enlargement factor). The average villous length was calculated from six measurements for each sample.

With the Ladewig staining, a typical trichrome stain, fibrin appears bright crimson red (Figure 4). Even small amounts of fibrin were easily identified due to the high contrast with the surrounding blue-stained connective tissue, making automatic color recognition by the image analysis software NIS-Elements^®^, version 4.5 (Nikon, Chiyoda, Tokyo, Japan) straightforward and interpretation always clear. This method was used to quantify the fibrin content for each case. The surface area proportions of fibrin deposits in the entire image section were presented as percentages (%). Thus, each case yielded a quantitative value representing the amount of extravasal inflammatory fibrin material in the sample.

### 2.6. Statistical Analysis

Data were analyzed using IBM SPSS Statistics^®^ software (Armonk, NY, USA), Version 22 for Windows. The Kruskal–Wallis ANOVA with post hoc testing, Man-Whitney test and Bonferroni correction were used to assess significant differences and correlations between the measurements and various animal groups. *p*-values less than or equal to 0.001 are considered highly significant, up to 0.01 as moderately significant, up to 0.05 as minimally significant, and values greater than 0.05 are regarded as not significant. Descriptive statistics were compiled in tables. Boxplots, bar charts, and area charts were used to illustrate the results obtained.

## 3. Results

### 3.1. Patients, Their Condition and Lameness Duration

A total of 78 joint capsule samples were examined for this study. Seventy-one dogs were presented due to hind limb lameness. Five of these dogs were later presented again due to lameness on the opposite side. Consequently, a total of 76 samples were obtained from surgical procedures, while 2 samples were collected post-mortem.

A medially directed patellar luxation was diagnosed in 59 cases (74%), and cruciate ligament rupture in 11 cases (16%). In cases of patellar luxation (PL), the condition was unilateral in 28 cases (47.46%) and bilateral in 31 cases (52.54%). A special group consisted of 4 dogs (5.13%) with both medial patellar luxation and cruciate ligament rupture (PL + CCLR). The control group included 4 dogs (5.13%) with other conditions (2 distal femoral fractures and 2 cases euthanized due to ileus or age-related euthanasia).

In the 59 cases of patellar luxation, the severity of the luxation varied significantly. The most commonly diagnosed grades that led to surgical intervention were grades 2, 3, and 4, with the diagnosis often depending on the examiner. In terms of frequency, 18 dogs (30.51%) were diagnosed with Grade 2 luxation; however, this grade was consistently identified by all examiners in only 6 cases (10.17%). Grade 3 was diagnosed 16 times (27.12%), while Grade 4 was identified and surgically treated in 4 cases (6.78%). Grade 1 patellar luxation was rarely observed, with only 2 instances recorded (3.39%). Additionally, 13 dogs (22.03%) exhibited luxation grades that fell between established categories: 3 dogs between Grades 1 and 2, 8 dogs between Grades 2 and 3, and 2 dogs between Grades 3 and 4.

Lameness duration was recorded in the different patients and summarized in the following Figure 5. In 20 patients (25.64%), the owners could not provide information on the duration of the lameness. Two control animals were not presented due to lameness but were included in the peracute lameness group.

### 3.2. Breed Group Frequencies

The dogs of different breeds and mixed breeds were divided into four weight groups (Figure 6).

In the PL group, Chihuahuas (*n* = 10) and Prague Rattlers (*n* = 2) were the most commonly represented breeds in the toy breed section, along with one Ruskiy Toy (*n* = 1). Among small breed dogs, mixed breeds (*n* = 6), Yorkshire Terriers (*n* = 5), and West Highland White Terriers (*n* = 3) were the most prevalent, with 14 additional cases from other breeds. In medium-sized breeds, mixed breeds (*n* = 5) and Shar-peis (*n* = 2) were most frequently observed, alongside seven other breeds. The large breed group included one German Shepherd, one Akita Inu, one American Staffordshire Terrier, and one mixed breed.

In the CCLR group, the toy breed category included one Chihuahua mix, while the small breed category featured one Shih Tzu. The medium-sized breed group consisted of one Tibetan Terrier, one Beagle, one Collie mix, and one mixed breed. The large breed category included two Boxers, one Caucasian Ovcharka mix, one Leonberger, and one Great Dane.

The PL + CCLR group, consisting of four individuals, equally represented the Yorkshire Terrier, Maltese, West Highland White Terrier, and a medium-sized mixed breed.

In the Control group (*n* = 4), breeds were equally represented by a Chihuahua, Miniature Poodle, a medium-sized mixed breed, and an American Staffordshire Terrier. The exact breed distribution is shown in Appendix A.

### 3.3. Age Characteristics of the Study Population

The age distribution of all patients ranged from 4 months to 15 years, with a mean age of 4.72 years and a median age of 4.00 years.

In the PL group, patient ages ranged from 0.5 years to 15 years, with a mean age of 4.22 years and a median age of 3.5 years. For the CCLR group, the minimum age was 2.5 years, and the maximum age was 12 years, with a mean age of 6.00 years and a median age of 11 years. In the PL + CCLR group, ages ranged from 4.5 years to 10 years, with a mean age of 7.3 years and a median age of 7.38 years. Finally, the Control group had patients ranging from 0.3 years to 15 years, with a mean age of 5.70 years and a median age of 3.75 years.

### 3.4. Body Weight Distribution

To better understand the distribution of body weight in the study population, the dogs were categorized into 10 weight classes, each with a range of 5 kg. This classification provides a clear overview of the weight distribution among the patients (Figure 7).

### 3.5. Gender Distribution and Characteristics

The study analyzed gender distribution among the canine patients, comprising 43 males and 35 females. Of the male dogs, 10 (29%) were neutered, while 16 females (37%) were spayed. No significant gender predisposition was identified for any specific condition.

Congenital patellar luxation was observed slightly more often in females (54.24%, *n* = 32) compared to males (45.76%, *n* = 27). Regarding cruciate ligament ruptures, 6 females and 5 males were affected, with one neutered dog in each group. In cases of concurrent patellar luxation and cruciate ligament rupture, 3 out of 4 dogs were female (75%), 2 of which were spayed, while the only male was neutered. Among the control group of four dogs, 2 were female and 2 were male, with one neutered individual. Overall, the findings suggest a balanced gender distribution among the study population, with no statistically significant gender differences in the prevalence of the conditions examined.

### 3.6. Histological Results

The histological stains were used to illustrate different structures in the 78 joint capsule biopsies, which will be presented in the following chapters.

#### 3.6.1. Characteristics of the Superficial Cell Layer

Two-thirds of the samples (*n* = 48, 61.54%) showed a single-layer superficial cell layer in at least one of the three measured areas (proximal, middle, and distal sections of the joint capsule samples). Among these, the superficial cell layer was single-layered in 8 cases (10.26%) across two of the three measurement areas, while in 3 samples (3.85%), the superficial cell layer consisted of a single cell layer in all three sections of the joint capsule (Figure 8B).

A multilayered superficial cell layer was defined as having more than 3 overlapping layers (Figure 8A). In 73.08% (*n* = 57) of the biopsies, the superficial cell layer was multilayered in at least one of the three measurement areas. The greatest variation in the number of superficial cells was observed in the middle section of the joint capsule, where the superficial cell layer of the stratum synoviale ranged from 1 to 15 layers. In contrast, the least variation was noted distally in the stratum synoviale, with between 1 and 8 superficial cell layers. The mean number of superficial cell layers in the proximal stratum synoviale was 3.88, which significantly differed from the mean of 2.66 in the middle section of the joint capsule (*p* = 0.004). Mean values from the other sections did not differ significantly from each other (Table 1).

The mean number of superficial cell layers in the proximal joint capsule section was similar among dogs with patellar luxation (mean = 4.03), those with cruciate ligament rupture, and those with both conditions. The mean value was lower in the control group. However, this difference was not statistically significant (*p* > 0.005) (Table 1).

##### Correlation Between Superficial Cell Layers and Disease

In the middle and distal joint capsule sections, greater fluctuations were observed between the individual groups, with the dogs in the control group showing the lowest mean value. However, there were no significant differences in the number of superficial cell layers between the control group and the other groups in these two measurement areas (*p* > 0.005). Dogs with both patellar luxation and an additional cranial cruciate ligament rupture, representing patients with a chronic-progressive disease course (longer duration of lameness), had the highest average number of superficial cell layers in the stratum synoviale. Conversely, dogs in the control group, characterized by acute lameness (no lameness to 1–2 days duration), had the lowest number of superficial cell layers (Figure 9).

##### Correlation Between the Superficial Cell Layers and Breed Groups

While differences in the mean number of superficial cell layers among the four breed groups were observed in the three joint capsule sections, these variations were not statistically significant. In toy, small, and medium-sized dog breeds, the middle section consistently showed fewer superficial cell layers on average compared to the proximal and distal sections. Large dog breeds were an exception to this pattern. For toy, small, and medium-sized breeds, the proximal section had the highest average number of superficial cell layers (Appendix A). Breed group had no significant influence on the number of superficial cell layers in any of the three joint capsule sections (*p* > 0.005).

##### Correlation Between Superficial Cell Layers and Duration of Lameness/Disease Progression

The mean number of superficial cell layers in the three joint capsule sections varied considerably with different durations of lameness; however, no statistically significant differences were observed (*p* > 0.005). On average, the middle superficial cell layer consistently exhibited the fewest cell layers compared to the superficial cell layers of the proximal and distal joint capsule sections across all forms of lameness progression. Notably, the smallest variation in values was observed during the peracute stage of lameness (Appendix A), with 75% of the control samples showing the least variation. Peracute lameness was associated with the lowest number of superficial cell layers compared to other durations.

##### Correlation Between Superficial Cell Layers and Age of the Patients

No correlation was found between the superficial cell layer and the age of the patients (*p* > 0.005). In none of the three joint capsule sections was there a direct relationship between the number of superficial cell layers and patient age. The middle superficial cell layer consistently exhibited the fewest cell layers across nearly all age groups (Appendix A).

#### 3.6.2. Surface Enlargement Factor—SEF

The Surface Enlargement Factor (SEF) quantifies villus formation on the inner surface of the joint capsule and serves as a grading system for this formation. The cases were categorized into three distinct groups based on the SEF values. A value of 1, observed in 3 cases (3.85%), indicates the absence of villi (Figure 10A). This value is derived from the ratio of the length of a smooth surface without villi to an equally long, parallel straight line. Values greater than 1 signify the presence of villi and are further subdivided: 41 cases (52.56%) had few or short villi (1 < SEF ≤ 2.5) (Figure 10B), while 26 cases (33.33%) exhibited many or long villi (2.5 < SEF ≤ 5) (Figure 10C). In 8 cases (10.26%), measurements could not be obtained due to the absence of the synovial layer, often resulting from tear-off or preparation damage. In total, 78 cases were assessed. The correlations between villus formation and patient signalment, along with data from the patients’ medical records, are discussed in the following section.

##### Correlation Between SEF and Disease

The mean values of the surface enlargement factors across different diseases show minimal variation. However, the group of dogs with PL (*n* = 52; 66.66%) exhibits a significantly greater variation in villus formation compared to the control group. Table 2 provides an overview of the mean values of the surface enlargement factors for the various diseases. In cases of patellar luxation (PL), some specimens displayed no villi (*n* = 2; 2.56%), while others had the highest counts of villi (*n* = 16; 20.51%). Most dogs with patellar luxation (*n* = 34; 43.59%) showed few or small villi. Similarly, among the 11 dogs with CCLR, small or few villi were the most commonly observed (*n* = 6; 7.69%). One case (1.28%) had no villi, and four cases exhibited very large or numerous villi (5.13%). In dogs with PL + CCLR, small or few villi were present in one case (1.28%), while tall or numerous villi were found in two cases (2.56%). All dogs in the comparison group (controls; *n* = 4; 5.13%) exclusively displayed tall or particularly numerous villi.

##### Correlation Between SEF and Luxation Grades of Patellar Luxation

There was no statistical correlation found between SEF and the severity of patellar luxation. For Grade 1–2 and Grade 2, the median SEF values were slightly above 2.0, with narrow ranges between 1.5 to 2.0 and 2.0 to 3.0, respectively. In grades ≥ 2, the median was just above 2.0, with a range between 1.0 and 3.0. Grade 3 showed a median SEF slightly above 2.0, but with a wider range from about 1.0 to 4.0. Grade 4 had a median value of 2.0, with a narrower range, approximately between 1.0 and 2.0 (Appendix A).

##### Correlation Between SEF and Disease Progression/Duration of Lameness

On average, cases with a subchronic course exhibited the largest villi (SEF = 2.92) and also showed the greatest variability, with a minimum value of 1 and a maximum of 4.94. The group with a peracute disease course displayed a similar range in the mean values of the surface enlargement factors, with minimum and maximum values of 1.09 and 4.86, respectively. In contrast, smaller villi were more commonly observed in cases of subacute and chronic-progressive lameness, while peracute, acute, and subchronic forms of lameness tended to develop larger villi (Appendix A and Appendix A).

##### Correlation Between SEF and Breed Groups

The mean values of the surface enlargement factors among the four breed groups showed minimal differences. Toy and small dog breeds exhibited, on average, villi of comparable size to those found in medium and large dog breeds. Notably, small dogs displayed the greatest variation in surface factors, with some individuals having the largest villi (SEF = 4.94) while others had no villi at all (SEF = 1.00). The variation, mean values, and extreme values for toy and large breeds were similar. Consequently, no proportional relationship between villus size and a dog’s body size could be established (*p* > 0.005) (Appendix A).

##### Correlation Between SEF and Age of the Patients

There is no correlation between the age of the patients and the surface enlargement factor. Young patients under one year of age have an average surface enlargement factor between 1.88 (small villi) and 2.99 (large villi). The mean values of the surface enlargement factors for older animals mostly lie between these two values. Therefore, no influence of age on the formation of villi could be demonstrated (*p* > 0.005). Appendix A illustrates the irregular distribution of the surface factors with respect to the age of the individual patients.

##### Correlation Between Villous Surface and Number of Superficial Cell Layers

In terms of their apical closure by the covering cells, no significant correlation could be established between the degree of villus formation and the number of covering cell layers in the three sections of the joint capsule. Apical closure refers to the sealing of the tips of the villi by the covering cells, which is important for maintaining the integrity of the joint capsule. Consequently, small villi can have as many or as few covering cell layers as large villi.

### 3.7. Absence of the Middle Stratum Subsynoviale

During the histomorphological examinations, it was observed that the stratum subsynoviale was absent in the middle section of the stifle joint capsule samples in 59 out of 78 cases (75.64%). As a result, the stratum synoviale was directly adjacent to the stratum fibrosum. This phenomenon was not observed in the proximal and distal sections of the joint capsule, where all three layers were always present (Figure 11A,B).

#### 3.7.1. Correlation Between Breed and Absence of the Stratum Subsynoviale in the Middle Section

The frequency of the presence of the stratum subsynoviale in the middle section of the stifle joint capsule was evaluated based on breed group. The middle section of the stratum subsynoviale was absent in nearly three-quarters of cases across toy, small, medium, and large dogs. No significant correlation between breed and the absence of the middle stratum subsynoviale was found (*p* > 0.005) (Appendix A).

#### 3.7.2. Correlation Between Age and Absence of the Stratum Subsynoviale in the Middle Section

In the age range of 0.3 to 1.0 years, the stratum subsynoviale was primarily present, although in limited samples. From 1.0 to 4.5 years, there was a notable increase in samples where this stratum was absent, leading to a decrease in its presence. By the age of 5.0 years, it was predominantly absent, with only a few instances noted at 5.5, 8.0, and 10.0 years. Overall, the data indicated that the absence of the stratum subsynoviale became more common with increasing age, particularly between 2.0 and 7.0 years. However, despite these findings, no significant correlation between age and the absence of the stratum subsynoviale was observed (*p* > 0.005) (Appendix A).

#### 3.7.3. Correlation Between Weight Classes and Absence of the Stratum Subsynoviale in the Middle Section

No correlation was found between the absence of the stratum subsynoviale in the middle section of the joint capsule and weight classes (*p* > 0.005). In all weight classes except for the heaviest (>45 kg), the stratum subsynoviale was predominantly absent in the middle section of the stifle joint capsule. In the lowest weight class (0–5 kg), this layer was absent in 74.07% of cases (*n* = 20), while in the next heavier class (5–10 kg), it was absent in 84.21% (*n* = 16). In the heavier weight classes, the ratio of absence to presence of the middle stratum subsynoviale changed to 2:1 for the 10–15 kg class and 7:1 for the 15–20 kg class. For the weight classes of 25–30 kg, 30–35 kg, and 35–40 kg, there was only one case in each class, with the middle stratum subsynoviale absent in all three instances. In the 40–45 kg weight class, the ratio of absence to presence of the middle stratum subsynoviale was again confirmed at 3:1 (Appendix A).

#### 3.7.4. Correlation Between Disease and Absence of the Stratum Subsynoviale in the Middle Section

A direct connection between the synovial layer and the fibrous layer in the middle section of the stifle joint capsule was observed in 81.36% (*n* = 44) of cases with PL and 63.64% (*n* = 7) of cases with CCLR. The condition of patellar luxation is significantly more frequently associated with the absence of the subsynovial layer compared to other diseases (*p* = 0.001) (Figure 12). Conversely, the subsynovial layer was present between the synovial and fibrous layers in only 18.64% (*n* = 11) of cases with PL and 36.36% (*n* = 4) of those with CCLR. In the group with both patellar luxation and cruciate ligament rupture, as well as in the control group, 50% of cases lacked a middle subsynovial layer.

#### 3.7.5. Correlation Between Absence of the Stratum Subsynoviale in the Middle Section and Duration of Lameness

In every stage of the examined lameness cases, significantly more instances without a middle subsynovial layer were recorded than those with this layer present. In cases of chronic-progressive lameness, the number of cases was highest in all groups with 23 cases (88.46%) (Appendix A). Thus, cases with a chronic-progressive course were significantly more frequently associated with the absence of the middle subsynovial layer than other progression forms (*p*-value = 0.001). In cases of acute lameness, this value was 85.71% (*n* = 6), whereas in cases of peracute and subacute lameness, it was 62.50% (*n* = 5 each). The group of subchronic lameness consisted exclusively of cases without this layer (*n* = 7, 100%). In the control group dogs, which were not presented due to lameness, the middle subsynovial layer was present in all cases.

### 3.8. Correlation of Thickness Among the Proximal, Middle, and Distal Sections of the Stifle Joint Capsule

The thicknesses of the three sections of the stifle joint capsule—proximal, middle, and distal—show a significant correlation with one another. A linear relationship among the four breed groups is illustrated in Appendix A. The Spearman correlation coefficients range from 0.327 to 0.536, indicating a strong correlation with a *p*-value of 0.001.

### 3.9. Correlation Between Breed Groups and Thickness of the Proximal, Middle, and Distal Sections of the Stifle Joint Capsule

Non-parametric Kruskal–Wallis tests for independent samples revealed significant differences in the thickness of the proximal and middle sections of the stifle joint capsule in relation to breed groups:

In toy dogs (*n* = 15), the average thickness (in μm) of the proximal section (1714.97 ± 914.47) was 46.41% smaller than in large dogs (*n* = 10) (*p* = 0.045, Bonferroni). It was also 44.60% smaller than in medium-sized dogs (*n* = 19) (3095.42 ± 1318.84) (*p* = 0.036, Bonferroni) (Figure 13).

Toy dogs had a significantly thinner middle section stifle joint capsule compared to all other breed groups. This data suggests that the size of the dog (toy, medium, large) significantly affects the thickness of the stifle joint capsule, particularly in the proximal and middle sections. Toy breeds tend to have thinner joint capsules compared to larger breeds.

Significant differences in thickness were observed among the four breed groups (Appendix A). Toy dogs exhibited significantly thinner layers in the proximal, middle, and distal sections of the stratum fibrosum compared to medium-sized dogs, with percentage differences of 42.70%, 68.03%, and 49.02%, respectively. The *p*-values indicate statistical significance (0.049, 0.006, and 0.048). In the proximal section of the stratum subsynoviale, both toy dogs and small dogs were found to be 66.85% and 65.73% thinner, respectively, when compared to large dogs (*p*-values of 0.002 and 0.001). The middle section of the stratum synoviale also showed significant thinning in toy dogs (63.96% thinner) compared to large dogs (*p* = 0.009), and in medium-sized dogs compared to large dogs (53.76% thinner, *p* = 0.021). Overall, these results suggest that smaller dog breeds tend to have significantly thinner tissue layers compared to larger breeds across various sections.

### 3.10. Correlation Between Disease and Thickness of the Stratum Synoviale

Significant differences in thickness were observed exclusively in the middle section of the stratum synoviale with respect to the different types of diseases (Figure 14). Dogs with patellar luxation exhibited a significantly thinner middle section compared to those with cruciate ligament rupture and the control group. Specifically, dogs with PL (*n* = 59) had a middle section that was 55.17% thinner than that of dogs with CCLR (*n* = 11) (*p* = 0.009, Bonferroni). Additionally, when compared to the controls (*n* = 4), dogs with PL had a middle section that was 64.65% thinner (*p* = 0.021, Bonferroni).

### 3.11. Comparison of Proximal and Middle Sections of the Stratum Fibrosum and Distal Section of the Stratum Synoviale Thickness in Unilateral vs. Bilateral Patellar Luxation

The average thicknesses of the proximal and middle sections of the stratum fibrosum, as well as the distal section of the stratum synoviale, were significantly different between cases of unilateral and bilateral patellar luxation. All three layer sections were significantly thinner in cases of bilateral patellar luxation (*n* = 31) compared to those with unilateral patellar luxation (*n* = 28).

Regarding the stratum fibrosum, it was, on average, 23.00% thinner in the proximal section (*p*-value = 0.013, Mann–Whitney U) and 45.13% thinner in the middle section (*p*-value = 0.017, Mann–Whitney U) in cases of bilateral patellar luxation compared to unilateral patellar luxation (Figure 15).

### 3.12. Middle Section of the Stratum Fibrosum and Patellar Luxation Grades

Significant differences were found in the average thickness of the middle section of the stratum fibrosum between dogs with patellar luxation grade 1 (*n* = 2) and those with grade 2 or higher (*n* = 20). The middle section was, on average, 71.92% thinner in dogs with more severe grades of patellar luxation compared to those with grade 1 (*p* = 0.036, Gabriel) (Figure 16).

### 3.13. Distal Section of the Stratum Subsynoviale and Degree of Villous Formation on the Inner Joint Surface

Cases without villi on the inner joint surface show a significantly thinner stratum subsynoviale in the distal section compared to those with abundant villi (*p*-value = 0.030, LSD). On average, this layer is 66.85% thinner in cases without villi (*n* = 3) than in those with numerous or prominent villi (*n* = 26) (Appendix A).

### 3.14. Extracellular Fibrin Deposits in the Stifle Joint Capsule Connective Tissue

The area proportions of extracellular fibrin deposits in the connective tissue of the stifle joint capsule showed neither significant differences nor correlations with breed groups, age, sex, weight, type of disease, duration of lameness or degree of villous formation. Out of 78 cases, 13 (16.66%) were free of fibrin deposits. This included 6 dogs with patellar luxation, 5 with cruciate ligament rupture, and 1 with both patellar luxation and cruciate ligament rupture.

When present, fibrin deposits could be found disseminated throughout all three layers and in the proximal, middle, and distal sections of the stifle joint capsule (Figure 4). The area proportions of fibrin ranged from 0.01% to 29.85% of the total area of the stifle joint capsule specimen. In 82.22% of cases with fibrin deposits, the proportion was 10% or less; in 13.51% of cases, it ranged from 10% to 20%; and in 5.41% of cases, it fell between 20% and 30%.

## 4. Discussion

Congenital patellar luxation (PL) and cranial cruciate ligament rupture (CCLR) are common hereditary musculoskeletal disorders in dogs. Patellar luxation, usually medial, results from skeletal malformations and soft tissue abnormalities, including joint capsule laxity. CCLR is associated with factors like an excessive tibial plateau angle, obesity, age, breed predisposition, and ligament degeneration [1,2,3,4]. While skeletal changes in PL are well documented, the role of soft tissue structures, particularly the stifle joint capsule, remains insufficiently studied.

Human studies on connective tissue disorders, such as those associated with COL6A1 mutations [15], suggest that similar mechanisms may also apply to dogs with PL. Congenital connective tissue disorders in dogs often mirror these human conditions [15,16] indicating a possible link between genetic predispositions and structural changes in the stifle joint capsule. However, it remains uncertain whether structural changes in the stifle joint capsule, including soft tissue laxity, represent primary defects driving PL or secondary adaptations resulting from skeletal malalignment and altered mechanical forces. Clarifying this relationship is crucial for a comprehensive understanding of the pathophysiology of PL and CCLR.

To explore this further, our study analyzed the thickness of joint capsule layers, superficial cell density in the stratum synoviale, villous formation, and fibrin content. These findings provide valuable insight into the relationship between skeletal deformities, PL or CCLR, and the histological changes observed in the stifle joint capsule.

Recently, new anatomical structures have been identified, such as a novel layer of elastic fibers between the synovial and fibrous membranes in the human shoulder [5]. These findings highlight the importance of reexamining the histomorphological characteristics of joint capsules. Consequently, the present study aims to reassess the structure of both healthy and affected canine stifle joint capsules, particularly in cases of PL and CCLR, to provide a more comprehensive understanding of their morphological differences.

Our study aims to fill a gap in current research by providing comprehensive data on stifle joint capsule thickness and associated histological changes in dogs with congenital patellar luxation and/or CCLR. By understanding these structural alterations, we hope to shed light on the etiology of these orthopedic diseases and inform breeding practices to reduce its prevalence in canine populations. The inclusion of dogs with distal femoral fractures as controls was based on the rationale that these patients required capsular imbrication, which involved the removal of a small portion of the joint capsule as part of their treatment. Additionally, distal femoral fractures are generally acute orthopedic conditions that are unlikely to have caused significant alterations to the joint capsule’s structural integrity, making these dogs suitable for comparison. The use of completely healthy individuals or those with unrelated health issues as controls is limited due to ethical considerations and the challenges associated with acquiring such samples (e.g., following euthanasia).

All PL patients included in this study exhibited medial patellar luxation, which is the most common form of patellar luxation, as demonstrated in previous studies [3,17,18]. Consequently, lateral capsular imbrication was performed to tighten the joint capsule on the lateral side, also for CCLR cases. In terms of sample collection, the stifle joint, whether right or left, was routinely exposed laterally via a parapatellar approach [12]. It is important to note that the tissue samples of the joint capsule were exclusively examined from the lateral parapatellar region for histomorphological analysis of texture and collagen patterns. This raises the possibility that the capsule may exhibit different structural characteristics medially, not only in terms of sections but also in its layering. Additionally, it is worth considering that laterally directed patellar luxations might exhibit different structural characteristics. In these cases, the lateral displacement of the patella could lead to elongation of the joint capsule medially rather than laterally.

This surgical technique enhances stifle joint stability by overlapping or suturing the lateral joint capsule to counteract the medial displacement of the patella [12]. This condition results in significant lateral elongation of the joint capsule, particularly in grades 3 and 4, with a corresponding medial narrowing. These observations suggest potential structural differences between lateral and medial aspects of the joint capsule. This hypothesis is supported by research on other joints. For instance, Bey et al. [19] demonstrated regional thickness variations in the human shoulder joint capsule. While not directly comparable, studies on other joints provide insights into potential capsular adaptations. Germscheid et al. [20] observed increased myofibroblast presence in the anterior section of traumatized human elbow joint capsules compared to healthy tissue. Similarly, Hecht et al. [21] reported increased alpha-actin expression from smooth muscle cells in sheep knee joint capsules, noting differential thermal effects between lateral and medial aspects after laser irradiation, which resulted in altered layer and collagen patterns. These findings collectively suggest that joint capsules may exhibit regional structural and cellular differences, which could be relevant to understanding the pathophysiology of patellar luxation in dogs.

In our study, the breeds most affected by PL among purebreds were Chihuahuas (15.39%), Yorkshire Terriers (8.97%), and West Highland White Terriers (5.13%). These findings are consistent with existing literature [3,18,22], which indicates that congenital PL predominantly affects small breed dogs. In the CCLR group, affected toy and small breeds included one Chihuahua mix and one Shih Tzu, while medium-sized breeds consisted of a Tibetan Terrier, a Beagle, a Collie mix, and a mixed breed. Larger breeds in this group included Boxers (*n* = 2), a Caucasian Ovcharka mix, a Leonberger, and a Great Dane. The PL + CCLR group comprised four individuals: a Yorkshire Terrier, a Maltese, a West Highland White Terrier, and a medium-sized mixed breed, suggesting that both small and medium-sized breeds can be predisposed to this combination of conditions [3,18,22].

The average age of dogs with PL in this study was 4.22 years, which aligns with the findings of Isaka et al. and Wangdee et al. [23,24]. This suggests that PL is commonly diagnosed in relatively young dogs, highlighting the importance of early detection and intervention. Conversely, the average age of dogs with CCLR was 6 years, consistent with reports by Gatineau et al. and Thompson et al. [25,26]. This age difference may indicate that CCLR are more likely to occur as dogs age, possibly due to degenerative changes on the ligament over time [3]. Among the small subset of patients (*n* = 4) with both patellar luxation and cruciate ligament rupture, the average age was 7.8 years, mirroring the findings of Campbell et al. [27] and closely aligning with the 7.38 years reported by Candela Andrade et al. [3]. This overlap in age suggests that dogs with pre-existing PL may be at increased risk for developing additional joint issues, emphasizing the need for comprehensive management of these patients.

In terms of sex distribution, females were slightly more frequently affected by PL, comprising 54.24% (*n* = 32) of cases compared to 45.76% (*n* = 27) in males. This trend is consistent with previous studies [1,18,21,22,28], although it contrasts with Singleton’s findings [10]. The higher incidence of PL in females may be linked to estrogen’s role in cartilage cell proliferation, as suggested by Gustaffson and Priester [22,28]. However, it is essential to recognize that other studies have reported equal occurrences in both sexes, indicating that the relationship between sex and patellar luxation remains a topic of debate [3]. Regarding CCLR, the distribution was relatively balanced, with six females and five males affected, including one neutered dog in each group, in line with previous literature [3].

The superficial cell layer of the synovial stratum consists of two overlapping layers in healthy human joints [29,30]. Sagiroglu’s studies [31] on 24 Kangal mixed breeds revealed age-dependent variations in the stifle joint capsule’s superficial cell layers. Specifically, young dogs (0–3 months) typically have 1–2 layers, while dogs between 3.5–6 months and older dogs (7 months to 6 years) exhibit 3–5 or 2–6 overlapping layers, respectively. Superficial cells are pleomorphic, varying in size, shape, number, and orientation, influenced by the adjacent subsynovial stratum [30,32]. These cells can differentiate into osteoblasts, adipocytes, and chondrocytes, identifiable by specific surface markers [33]. Factors such as age, joint health, and location significantly affect superficial cell morphology [30]. Bronner and Farach-Carson [34] highlight the critical role of the surface cell layer in “controlling the environment of the joint,” while Moskalewski et al. [29] suggest that due to the heterogeneous properties of these cells, the superficial cell layer should be regarded as an independent organ, distinct from the fiber-rich joint capsule.

These characteristics are reflected in the variable numbers of superficial cell layers observed in this study. On average, 3 to 4 layers were present in the proximal and distal sections, while the middle section often exhibited a single layer. In 73.08% (*n* = 57) of cases, at least one section (proximal, middle, or distal) had a multilayered superficial cell layer with a minimum of 3 layers. Wondratschek [35] reported similar findings, noting that 50% of dogs with stifle osteoarthritis had multilayered superficial cell layers. Among the dogs in this study, those with chronic-progressive conditions (lameness duration > 90 days) had the most superficial cell layers, particularly in cases of concurrent patellar luxation and cruciate ligament rupture. Dogs PL had approximately a similar number of superficial cell layers as those with CCLR. In contrast, control dogs with the shortest duration of lameness (no lameness to 1–2 days) exhibited the fewest layers, which aligns with findings from Manunta et al. [36]. In their research, Manunta et al. [36] observed that in human athletes, an increase in multilayering of the surface cell layer was associated with hyperplasia, exudative characteristics, and infiltration by inflammatory cells. This suggests that the response of the surface cell layer may vary depending on the underlying condition and duration of lameness, indicating a potential link between joint health and the cellular response in both dogs and humans. This similarity between PL and CCLR cases likely reflects the influence of chronic joint instability rather than the degree of radiographic osteoarthritis. Even though PL is often considered less degenerative, prolonged abnormal loading and repetitive soft-tissue strain can still create enough synovial irritation to elicit a hyperplastic response comparable to that seen in CCLR.

The degree of villus formation on the inner joint surface was assessed using the surface enlargement factor (SEF), with higher values indicating larger or more numerous villi in the joint capsule samples. Dogs with PL exhibited the greatest variations in villus formation (1.00–4.94), while all control animals exhibited predominantly high or numerous villi (2.64–4.86), resulting in a mean value of 3.54 that significantly differed from the chronic joint pathologies (2.46). This variability in SEF values, particularly in dogs with PL, may be influenced by chronic synovial effusion. Chronic capsular distention caused by persistent effusion could stimulate villus formation, contributing to the histological changes observed in this study. In contrast, the narrower SEF range observed in dogs with CCLR (1.00–3.82) likely reflects the acute nature of the condition, which minimizes chronic remodeling of the joint capsule. Comparable patterns of synovial villus behavior have also been described in other species, including humans and horses, where chronic joint instability leads to progressive synovial surface remodeling [30,37]. These findings underscore the pivotal role that the chronicity of joint instability plays in villus development, emphasizing the importance of future research on this relationship.

Bertone’s [37] results showed that chronic joint diseases lead to increasingly blunt and shorter synovial villi, often taking on a club-shaped appearance due to reperfusion-induced ischemia and resultant hypoxia, particularly affecting the tips of the villi. Other observations have been reported in dogs with CCLR, where synovial villi ranged from filamentous to club-shaped, contributing to the hyperplastic transformation of the inner joint capsule surface [38]. Wondratschek [35] also described the diversity of synovial villi in dogs with osteoarthritic changes, reporting that 96% exhibited villous hyperplasia with a wide range of shapes and sizes, sometimes even within a single sample. Our study confirmed that small-breed dogs exhibited significant variation in villus formation, with some displaying the largest villi while others had none. Notably, cases without villi had a significantly thinner distal subsynovial stratum compared to those with abundant villi, suggesting a possible relationship between subsynovial structure and villus development. Comparable patterns of synovial surface remodeling have also been described in humans, where chronic joint disease and mechanical irritation lead to alterations in villous morphology as well as changes in the synovial lining and subsynovial connective tissue [30]. These results warrant further investigation into the role of chronicity in villous development within the joint capsule.

The absence of the stratum subsynoviale has not been explicitly documented in the existing literature. However, in 59 out of 78 cases (75.64%), the stratum subsynoviale in the middle section of the stifle joint was undetectable, representing a significant finding. In these cases, the stratum synoviale directly adjoined the stratum fibrosum, while all three layers were detectable in the proximal and distal sections of the same sample. Notably, dogs with PL were significantly more likely to lack the stratum subsynoviale in the middle section, with 81.36% (*n* = 44) affected, compared to 63.64% (*n* = 7) of dogs with CCLR.

The absence of the stratum subsynoviale was significantly associated with the duration of lameness; in chronic-progressive cases (lameness duration > 90 days), 88.46% (*n* = 23) of the dogs were affected, indicating a stronger correlation with longer lameness durations. While the stratum fibrosum is typically regarded as the primary layer contributing to the mechanical stability of the joint capsule, the stratum subsynoviale may play an indirect but important role in maintaining tissue integrity. This layer provides structural support to the overlying stratum synoviale and facilitates nutrient transport, cellular turnover, and vascularization, which are essential for synovial function and joint health. Its absence in the middle section may impair these processes, potentially leading to degenerative changes in the synovial layer and reduced adaptability of the joint capsule to mechanical forces. This raises the question of whether the stratum subsynoviale in these 59 cases with PL may have existed previously but underwent changes that rendered it indistinguishable from the adjacent stratum fibrosum. Chronic synovial effusion and prolonged joint instability, common in PL, could contribute to fibrotic remodeling and a loss of clear demarcation between these layers, as the tissue adapts to abnormal loading conditions. Notably, some recent literature does not acknowledge the stratum subsynoviale, suggesting that the joint capsule consists solely of an inner stratum synoviale and an outer stratum fibrosum [39,40,41,42,43].

Thickness measurements in this study revealed significant variation in the stifle joint capsule across the proximal, middle, and distal sections, with these measurements showing strong correlation with one another. This aligns with the findings of Bey et al. and Wagner et al. [19,40], who highlighted regional variations in joint capsule thickness as adaptations to loading conditions. The joint capsule, functioning as a limiting structure for synovial joint motion, undergoes continuous histomorphological remodeling in response to mechanical loads. Ralphs and Benjamin [44] noted that the joint capsule thickens at its bony attachments, which was corroborated here, as the proximal section was thicker than the middle section in 93.60% (*n* = 73) of cases.

In this study, we observed that dogs with PL had a significantly thinner synovial layer in the middle section, reduced by 55.17% to 64.65% compared to the other groups (CCLR and controls, respectively). This variation may be influenced by the functional demands and mechanical adaptations of the joint. It is well known that the joint capsule thickness increases near its attachment sites, where it contains more vessels, fat cells, and synovial lining compared to the midsection. The thinner layers in dogs with PL may reflect reduced mechanical loading or functional disruption in these regions. Bilateral PL may also contribute to these microscopic changes, as it has been associated with thinner fibrous layers in the proximal and middle regions, as well as a thinner synovial layer in the distal region, compared to unilateral cases. This suggests that weight shifting to the healthy limb in unilateral luxation could mitigate joint capsule changes.

For dogs with CCLR, the increased synovial thickness observed could result from proliferative lymphoplasmacellular synovitis, consistent with previous studies [14,38,45]. In femur fracture cases within the control group, instability in the adjacent stifle joint may contribute to synovitis and subsequent thickening. These findings highlight the importance of accounting for regional variations in capsule thickness, of luxation, and normal synovial plication when interpreting joint capsule histomorphology. Higher grades of PL and associated capsule changes underscore the complex interplay between functional disruption, loading conditions, and compensatory mechanisms.

Significant differences in joint capsule thickness were observed among breed groups, with toy dogs showing a thinner proximal section compared to medium and large breeds, and a thinner middle section relative to all other groups.

Analysis of extracellular fibrin deposits in the stifle joint capsule revealed no significant differences or correlations with breed, age, sex, weight, type of disease, duration of lameness, or degree of villous formation. This suggests that fibrin presence may reflect a generalized response within the joint capsule, rather than being tied to specific individual characteristics or conditions. Insights from Oberbauer et al. [16] and Hildebrandt et al. [6] indicate that genetic factors and trauma responses may collectively shape the histopathological landscape of the joint capsule in conditions like PL and CCLR. Further research could explore the implications of these fibrin deposits in relation to joint function and recovery, particularly in the context of surgical interventions.

This study has several limitations that should be acknowledged. Firstly, the absence of immunohistochemical staining to analyze collagen patterns within the stifle joint capsule is a significant limitation. Elastic fibers and specific collagen subtypes are critical for joint stability and mechanical adaptation, and their distribution and structural organization remain unexplored in this study. Future research incorporating immunohistochemical techniques could provide valuable insights into collagen patterns, particularly in dogs with patellar luxation compared to unaffected dogs. Additionally, biomechanical analyses to assess differences in the mechanical properties of collagen fibers between affected and unaffected dogs would complement histological findings and enhance our understanding of joint capsule adaptations.

A further limitation of this study concerns the grading of patellar luxation. Several cases received different grades when evaluated by different veterinarians, and these animals were therefore assigned to an intermediate category. Although this approach reduced inter-observer discrepancies, it may still have affected the ability to detect clear grade-specific histological differences. Future studies that incorporate detailed assessments of structural changes in the surrounding hard tissues—such as femoral trochlear morphology, tibial alignment, or rotational deformities—could help define the patellar luxation grade more precisely. Integrating these objective skeletal parameters may reduce grading ambiguity and, in turn, minimize the dilution of histological distinctions between severity categories.

Control samples were obtained from dogs with distal femoral fractures or from dogs euthanized for unrelated health conditions, with owner consent. Obtaining completely healthy stifle joint capsules is ethically challenging, as owners are generally reluctant to allow tissue collection after euthanasia. Ideally, future studies should include samples from truly healthy dogs—and, where possible, from animals within a comparable age range—to establish a more accurate baseline. However, these ethical and practical constraints make such sampling difficult. In this context, collaboration with other institutions may help increase access to suitable and better-matched control material. These limitations underscore the need for further research to address these gaps and build upon the findings of the present study.

## 5. Conclusions

This study demonstrated distinct histological differences in the stifle joint capsule among dogs with patellar luxation (PL), cranial cruciate ligament rupture (CCLR), and control animals. Dogs with PL and CCLR exhibited an increased number of superficial synovial cell layers compared to controls. Chronic cases showed reduced villous formation as reflected by lower SEF values. Absence or marked reduction of the stratum subsynoviale was associated with PL and prolonged lameness. In PL cases, the stratum synoviale was frequently absent, whereas CCLR cases showed an increase in overall capsular thickness. These findings indicate that PL and CCLR are associated with different patterns of histomorphological remodeling of the joint capsule, and that chronicity of disease influences these structural alterations.

## Figures and Tables

**Figure 1 animals-15-03545-f001:**
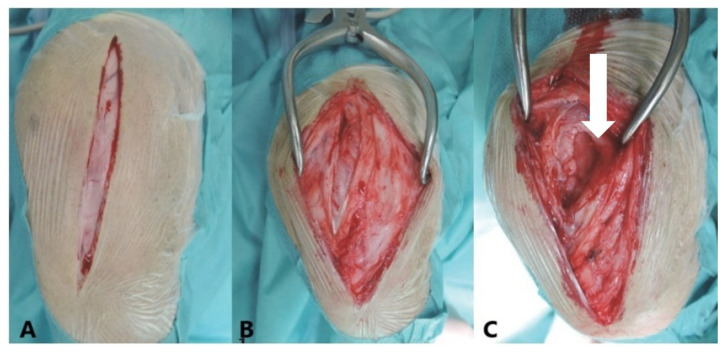
Lateral view of a left canine stifle joint illustrating a lateral parapatellar approach. (**A**)—The skin is incised parapatellarly. (**B**)—The fascial tissue is opened parapatellarly. (**C**)—The joint capsule (arrow) is exposed and opened for sample removal and to facilitate a capsular imbrication procedure. Posterolateral tightening of the joint capsule helps realign the patella in cases of patellar luxation (PL), contributing to improved stability and joint function. In cranial cruciate ligament rupture (CCLR) cases, additional fascial imbrication is performed using the Meutstege technique to further enhance joint stabilization.

**Figure 2 animals-15-03545-f002:**
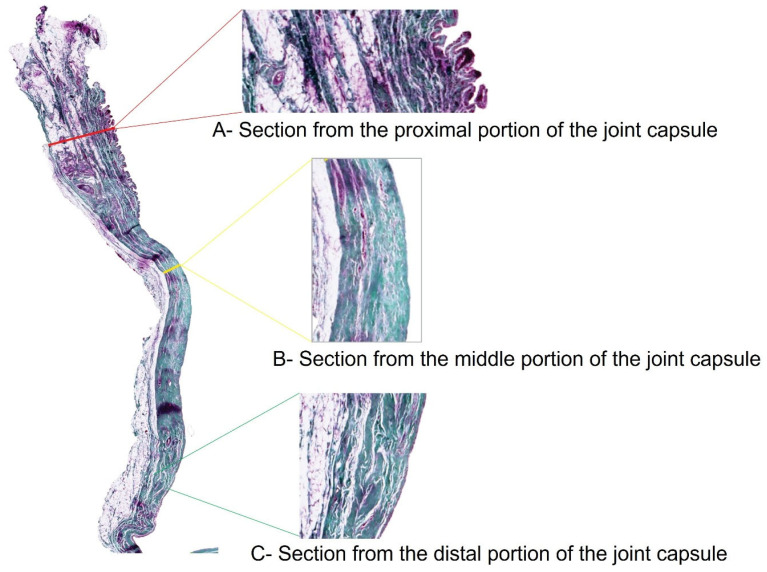
Stifle joint capsule sample from a dog with patellar luxation. Thickness measurements of the proximal, middle and distal sections. Staining according to Volkmann–Strauss. Magnification: 1.25× (whole joint capsule sample), 5× (individual sections).

**Figure 3 animals-15-03545-f003:**
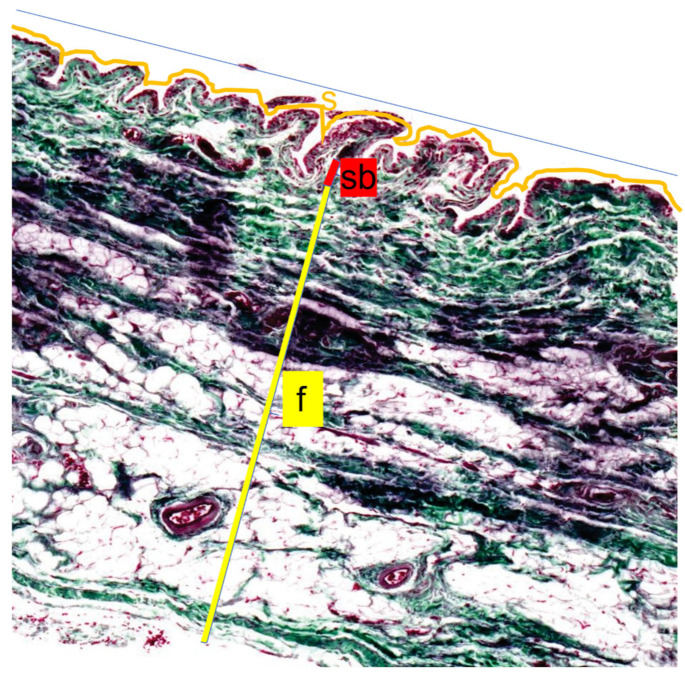
Length measurements of the inner joint surface. The Length measurements of the inner joint surface were performed by comparing the surface length of a villous section (ZL, orange line) with the corresponding straight reference line (G, blue line) to calculate the surface magnification factor (SMF = ZL/G). Thickness measurements were obtained for the three layers of the joint capsule: the Stratum synoviale (s), a cell-rich layer lining the joint cavity; the well-vascularized Stratum subsynoviale (sb); and the outer Stratum fibrosum (f), which contains abundant collagen fibers (yellow). Staining was performed using the Volkmann–Strauss method, and all measurements were assessed at 10× magnification.

**Figure 4 animals-15-03545-f004:**
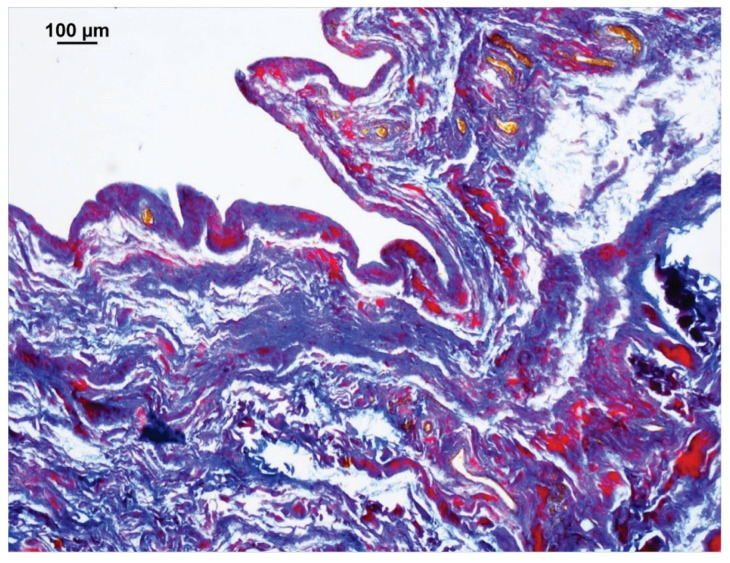
Section of a stifle joint capsule sample from a dog that was euthanized for an intestinal obstruction. The image analysis software NIS-Elements^®^ detected the purple to red-stained fibrin deposits. The software then calculated the percentage surface area of fibrin relative to the entire image. Connective tissue is stained blue, erythrocytes are yellow-orange. Staining method: Ladewig.

**Figure 5 animals-15-03545-f005:**
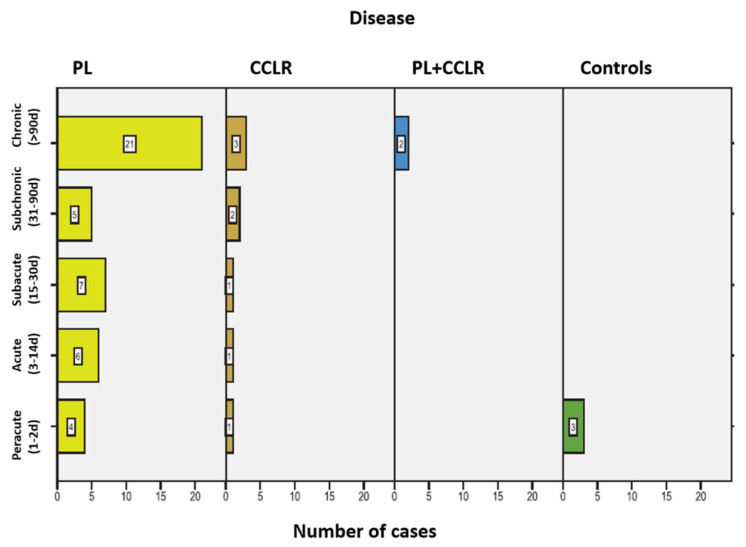
Frequency of lameness duration among the dogs in the study, including the associated conditions for each duration category. Abbreviations: Patellar luxation—PL; Cranial cruciate ligament rupture—CCLR.

**Figure 6 animals-15-03545-f006:**
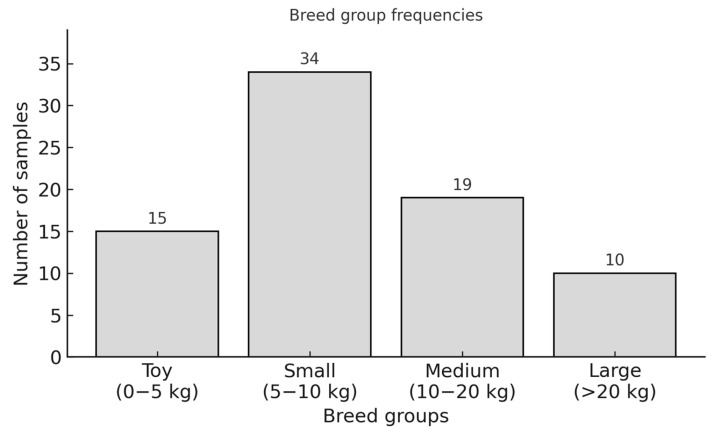
Breed group frequencies per size and weight.

**Figure 7 animals-15-03545-f007:**
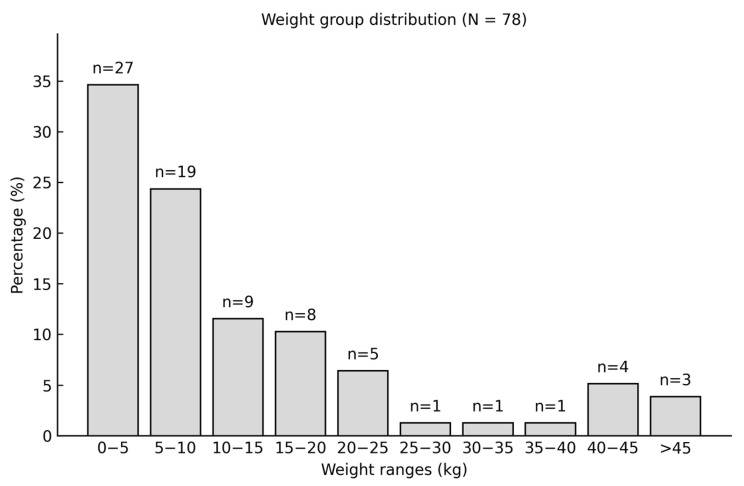
Body weight distribution and number of patients in each weight class in the study population.

**Figure 8 animals-15-03545-f008:**
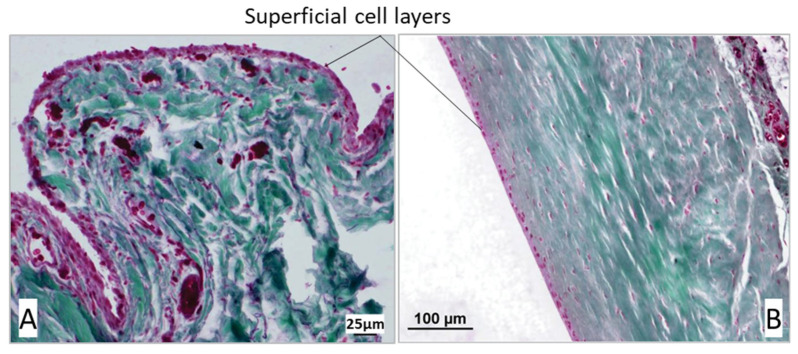
(**A**) Multilayered, oval-shaped surface cells in the proximal section of the joint capsule in a dog with cranial cruciate ligament rupture. (**B**) Single-layered, flat surface cells in the middle section of the joint capsule in a dog with patellar luxation. In the Volkmann–Strauss staining, the cytoplasm of the surface cells appears pink, whereas the nuclei show a dark red–violet coloration. Collagen fibers are stained green, and the white areas correspond to the lumen of the joint capsule or artifact spaces from tissue processing. Scale bars: A = 25 µm; B = 100 µm.

**Figure 9 animals-15-03545-f009:**
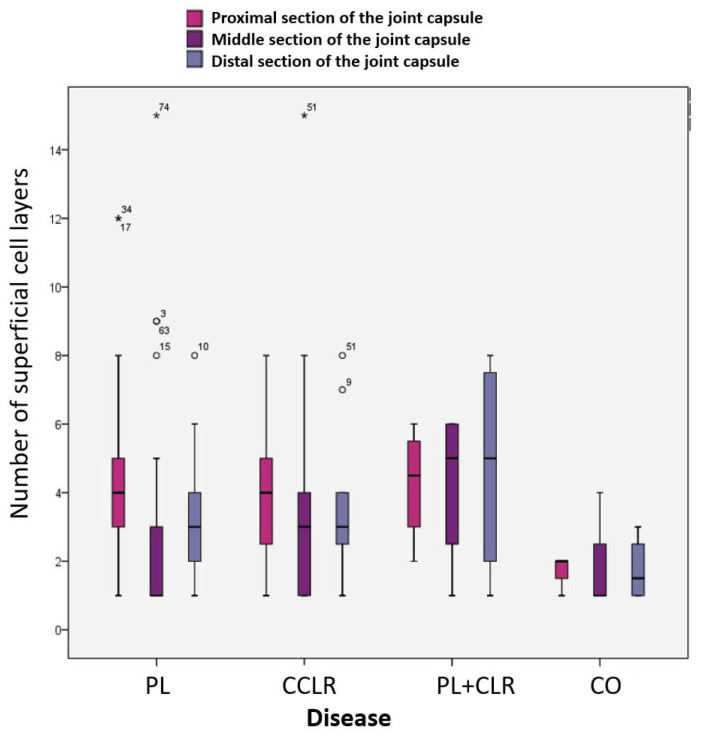
Number of superficial cell layers in the proximal, middle, and distal stratum synoviale in the four subject groups.

**Figure 10 animals-15-03545-f010:**
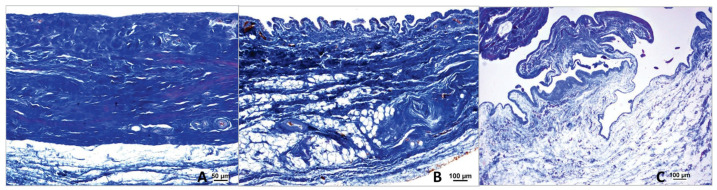
Different grades of villous formation based on the surface enlargement factor. (**A**) Absence of villi in the middle stifle joint capsule of a dog with patellar luxation; stained using the Ladewig method. (**B**) Small villi in the proximal stifle joint capsule of a dog with patellar luxation; stained using the Ladewig method. (**C**) Large villi in the middle stifle joint capsule of a dog with a femoral fracture; stained using the Ladewig method. In the Ladewig staining, collagen fibers are stained blue. Scale bars: A = 25 µm; B and C = 100 µm.

**Figure 11 animals-15-03545-f011:**
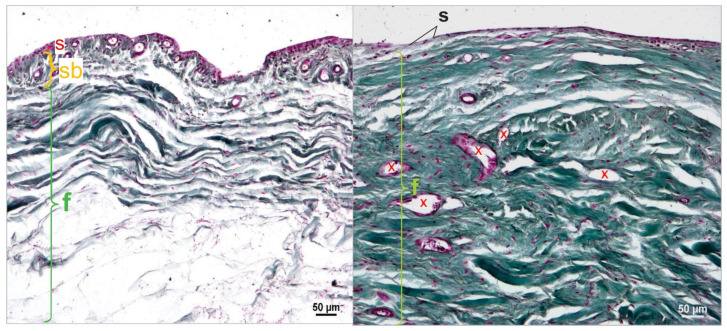
(**A**). Middle section of the stifle joint capsule; cruciate ligament rupture. The cell-rich *stratum synoviale* (s) (red) connects to the well-vascularized (x) *stratum subsynoviale* (sb) (orange), which transitions into the *stratum fibrosum* (f) (green), which contains collagen fibers (green). Staining according to Volkmann–Strauss. (**B**). Middle section of the stifle joint capsule; patellar luxation. The cell-poor, single-layered *stratum synoviale* (s), with flattened synoviocytes and fibroblasts, directly connects to the collagen-fiber-rich (green) and well-vascularized (x) *stratum fibrosum* (f) (green). Staining according to Volkmann–Strauss. In Volkmann–Strauss staining, collagen fibers are stained green, whereas the superficial cell layers of the *stratum synoviale* (s) and the epithelial cells of blood vessels appear violet.

**Figure 12 animals-15-03545-f012:**
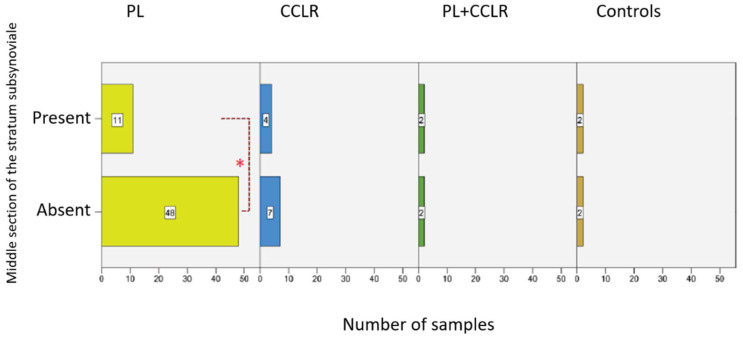
Number of cases with a present or absent subsynovial layer in the middle section of the stifle joint capsule for the examined diseases. Statistical significance was set at *p* < 0.05 and is indicated by an asterisk (*).

**Figure 13 animals-15-03545-f013:**
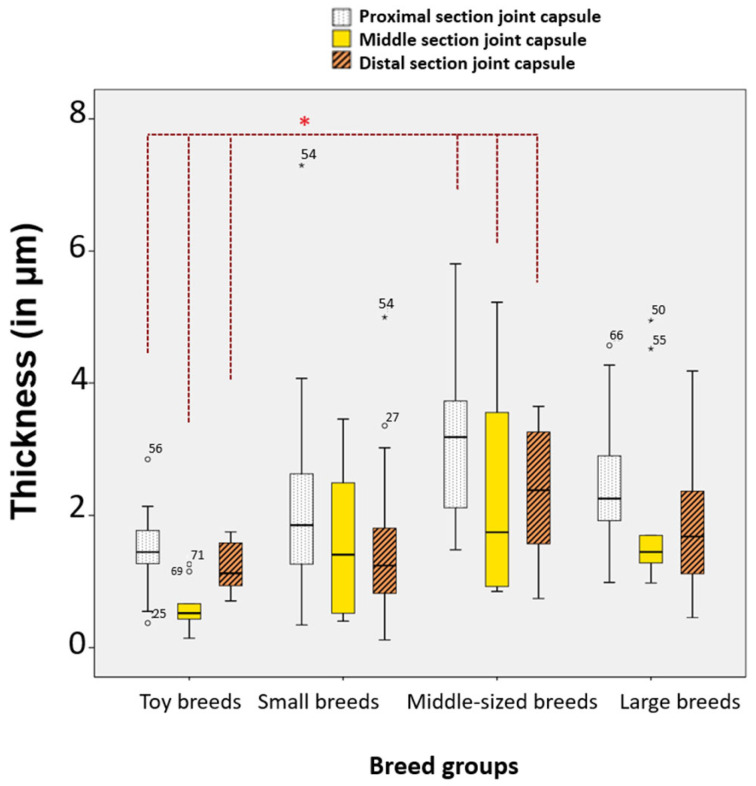
Significant differences in selected sections and layers of the stifle joint capsule among the four breed groups. Statistical significance was set at *p* < 0.05 and is indicated by an asterisk (*).

**Figure 14 animals-15-03545-f014:**
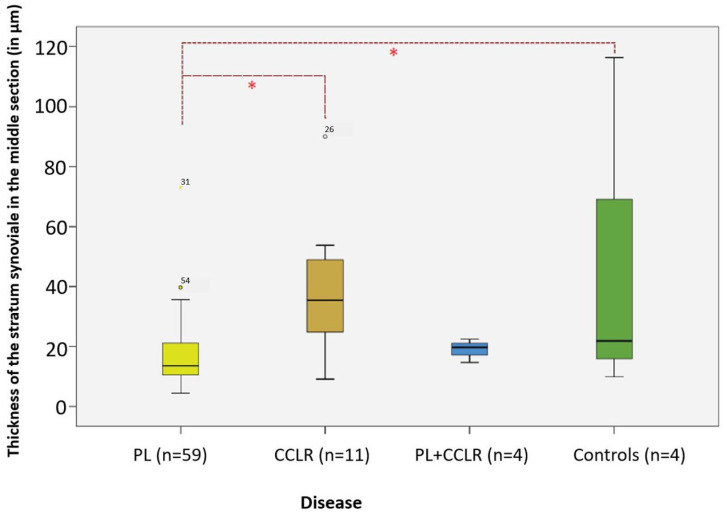
Thickness of the middle section of the stratum synoviale in the studied diseases compared to controls. Significant differences in thickness are highlighted in the analysis. Significance level: *p* < 0.05. Abbreviations: PL—patellar luxation; CCLR—cranial cruciate ligament rupture. Statistical significance was set at *p* < 0.05 and is indicated by an asterisk (*).

**Figure 15 animals-15-03545-f015:**
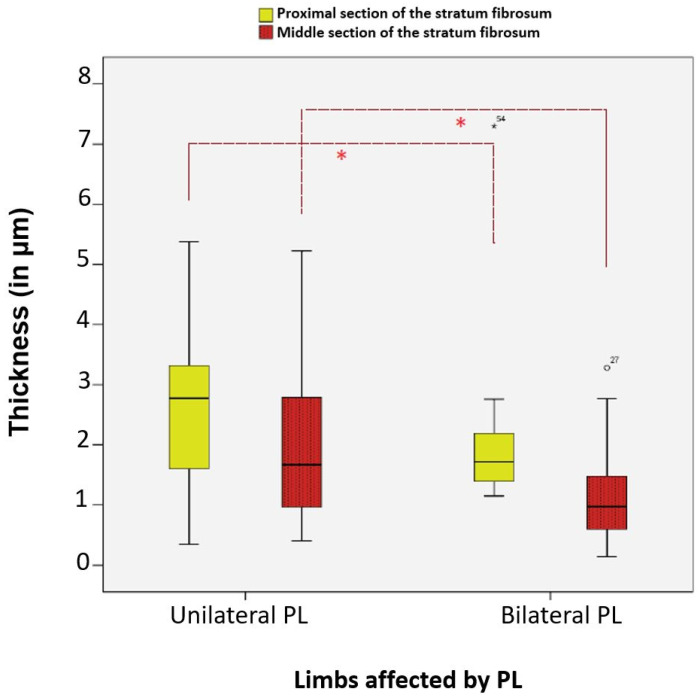
Significant differences in the average thickness of the proximal and middle sections of the stratum fibrosum in dogs with unilateral and bilateral patellar luxation. Statistical significance was set at *p* < 0.05 and is indicated by an asterisk (*).

**Figure 16 animals-15-03545-f016:**
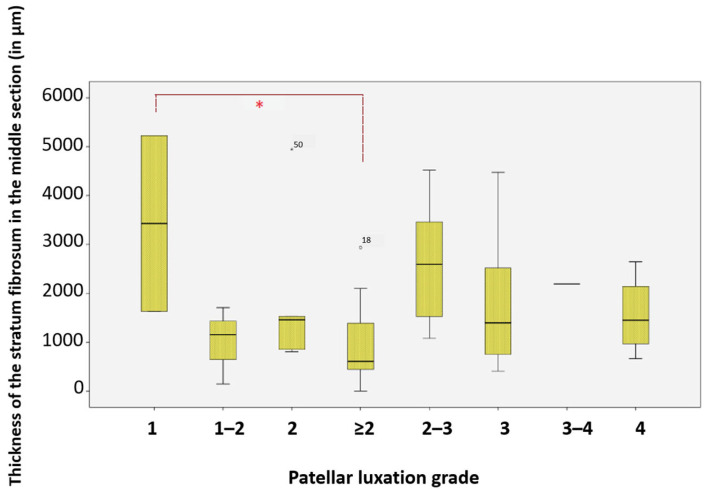
Thickness of the stratum fibrosum in the middle section at different patellar luxation grades, Significance (*) between grade 1 and grade ≥ 2. * Significance level: *p* < 0.05. For statistical analysis, grades II–IV were grouped into a single category (≥II) to represent clinically relevant instability and to improve statistical robustness, as individual comparison between grades did not reveal additional significance. Statistical significance was set at *p* < 0.05 and is indicated by an asterisk (*).

**Table 1 animals-15-03545-t001:** Number of superficial cell layers in the 3 areas of the joint capsule—Stratum synoviale. Abbreviations: PL—Patellar luxation; CCLR—Cruciate ligament rupture, CO—Control group.

Superficial Cell Layers	Proximal Stratum Synoviale	Middle Stratum Synoviale	Distal Stratum Synoviale
Mean ± SD	3.88 ± 2.19	2.66 ± 2.80	3.21 ± 1.71
PL	4.03 ± 2.27	2.39 ± 2.51	3.14 ± 1.44
CCLR	3.73 ± 2.05	3.91 ± 4.23	3.55 ± 2.21
PL + CCLR	4.25 ± 1.71	4.25 ± 2.36	4.75 ± 3.30
CO	1.75 ± 0.50	1.75 ± 1.50	1.75 ± 0.96

**Table 2 animals-15-03545-t002:** Mean values and other characteristics of surface enlargement factors for different diseases. Abbreviations: PL—patellar luxation; CCLR—cranial cruciate ligament rupture; CO—controls.

Surface Enlargement Factors						
Disease	N	Minimum	Maximum	Mean	SD	Median
PL	52	1.00	4.94	2.27	0.85	2.16
CCLR	11	1.00	3.82	2.33	0.83	2.18
PL + CCLR	3	1.14	3.35	2.38	1.13	2.66
CO	4	2.64	4.86	3.54	0.96	3.33
Total	70	1.00	4.94	2.35	0.89	2.22

## Data Availability

The datasets used and/or analyzed during the current study are available from the corresponding author upon reasonable request.

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
