# Peer review of "Structural and Histomorphological Evaluation of the Stifle Joint Capsule in Canine Congenital Patellar Luxation and Cranial Cruciate Ligament Rupture"

_animals, 2025, doi:10.3390/ani15243545_

Round 1
Reviewer 1 Report
Comments and Suggestions for Authors
Line 21 / 39: please explain/compare these statements
Line 44: strange selection of keywords, I would prefer PL, CCLR, histomorphology, stifle joint capsule, dog
Line 48: please provide a literature reference indicating hereditary cause of CCLR in dogs.
Line 84: why did you do a capsular imbrication in dogs with a distal femoral fracture?
Line 124ff: I would prefer to skip this paragraph, because it seems to be unrelated to the study topic.
Line 143: skip the comma. Set a point. Did you do the Meutstege procedure in all weight groups?
Line 299: Why?
Discussion: please shorten this chapter and focuse on the main topic of this study – the histological examination and its results. Skip the paragraphs about age / breed / sex / weight. Influence of these parameters seems to be low regarding your results.
Line 785: approximately
Please explain, why you think there were similar layers in PL and CCLR dogs. In my experience, CCLR causes severe osteoarthritis and -arthrosis whereas PL joints are mainly free from OA.
Line 800: Did your clinical evaluation of the PL joint really reveal chronic synovial effusion? How old were the CCLRs that SEF values were that different to PL dogs?
Please use the discussion to answer these questions, explain the context and compare to other joints / animals / humans.
Line 922: which ones? What is the take-home message eg. for a Chihuahua breeder?
Conclusions: Please use this section to conclude the results, but do not make suggestions or state the importance of further research. This is not a conclusion!!!!!
Figure 1: different
Figure 7: green / red or better black / grey?
Author Response
Reviewer 1
We thank the reviewer for the constructive and highly valuable comments provided on our manuscript. We greatly appreciate the time and expertise invested in evaluating our work. We are fully aware of the limitations inherent to our dataset, and we have carefully revised and expanded the Limitations section to ensure that readers can clearly understand these constraints and their implications. All corresponding clarifications and modifications have been incorporated into the revised manuscript.
Reviewer:
Line 21 / 39: please explain/compare these statements
New line 21-22: These findings suggest that ongoing joint instability is associated with distinct altera-tions of the joint capsule, particularly affecting the composition and organization of its layers.
New lines 38-31: Absence or marked reduction of the stratum subsynoviale correlated with PL and pro-longed lameness. In PL cases, the stratum synoviale was frequently absent, whereas CCLR cases exhibited an increase in overall capsular thickness
We thank the reviewer for pointing out the need for clearer terminology. We have revised the text to avoid ambiguous terms such as “weakening” or “thinning” and now describe the histological findings with anatomically accurate wording. Specifically, we report the absence or marked reduction of the stratum subsynoviale, which correlated with PL and prolonged lameness. Additionally, in PL cases the stratum synoviale was frequently absent, whereas CCLR cases showed an increase in overall capsular thickness based on direct histological measurements. These revisions more precisely reflect the observed changes and avoid any unintended interpretative bias.
Reviewer: Line 44: strange selection of keywords, I would prefer PL, CCLR, histomorphology, stifle joint capsule, dog.
Answer: Thank you for your comment. This is now updated to: Keywords: knee; dislocation; tissue; dog; histomorphology, PL, CCLR, stifle joint capsule
Reviewer: please provide a literature reference indicating hereditary cause of CCLR in dogs
Answer: Thank you for your remark. Literature has been added.
Reviewer:
Line 84: why did you do a capsular imbrication in dogs with a distal femoral fracture?
We thank the reviewer for this relevant comment. Capsular imbrication in dogs with distal femoral fractures was performed at the discretion of the responsible surgeon. In these cases, both dogs presented with a hemarthros associated with the trauma and therefore required an additional arthrotomy for joint inspection and lavage. Following fracture reduction and fixation, the surgeon considered that capsular imbrication would provide additional functional stability to the stifle joint during recovery. This decision reflected the surgeon’s clinical judgement at the time of surgery and was not related to the aims of the study.
Explanation in the manuscript added: Dogs with distal femoral fractures that required an arthrotomy due to hemarthrosis. New line 97
Reviewer:
Line 124: I would prefer to skip this paragraph, because it seems to be unrelated to the study topic.
Author Response:
We thank the reviewer for this comment. We understand the concern that this paragraph may seem unrelated to the main study topic. However, one of the key limitations of our work is the difficulty in obtaining a truly healthy, orthopedically disease-free control group. Because of this, it is important for the reader to understand where the control samples originated and under which clinical circumstances they were collected. This point was also raised by another reviewer, and we therefore believe that retaining the paragraph provides valuable context regarding the nature of the control material.
As discussed, obtaining completely healthy stifle joint capsules from dogs of comparable age and free of any systemic disease is ethically and practically challenging. For this reason, we would prefer to leave the paragraph as it is, as it helps clarify these constraints and supports the transparency of our methodology.
Reviewer: Line 143: skip the comma. Set a point. Did you do the Meutstege procedure in all weight groups?
Answer: Comma issue was corrected. Yes, Meutstege procedure was used for all weight groups.
This technique can be succesfully used in all dog weight groups according to Allgoewer et al. 2000 (Comparison of two intra-extra-articular stabilization techniques for treatment of rupture of anterior cruciate ligament in dogs. FLO (mod.) and Meutstege techniques.)
Reviewer: Line 299: Why?
Author Response:
We thank the reviewer for this comment. Patellar luxation grading in our study was performed by different veterinarians, and—as previously reported in the literature (e.g., Weber 1992, Morphologische Studie am Becken vom Papillon-Hunden unter Berücksichtigung von Faktoren zur Ätiologie der nichttraumatischen Patellaluxation nach medial)—interobserver variability in PL grading is a well-known issue. In cases where grading differed between clinicians, we assigned an intermediate grade to avoid misclassification, which we believe was the most conservative and transparent approach.
We have therefore added to the Discussion that future studies should incorporate objective assessments such as radiographic measurements or skeletal morphometric parameters to better quantify structural contributors to patellar luxation. This would help reduce grading variability and improve the precision of PL classification.
Reviewer:
Discussion:
please shorten this chapter and focuse on the main topic of this study – the histological examination and its results. Skip the paragraphs about age / breed / sex / weight. Influence of these parameters seems to be low regarding your results.
Answer: We thank the reviewer for this helpful suggestion. We acknowledge that in our dataset the influence of age, breed, sex, and weight on the histological outcomes was limited. However, these factors are extensively discussed in the literature as important contributors to the predisposition for both patellar luxation and cranial cruciate ligament rupture. For this reason, we believe that a brief contextualization of these parameters is valuable for readers who wish to understand how our findings relate to the broader epidemiological background of these conditions.
Since Animals does not impose a strict word limit, and given that these paragraphs help position our results within the existing body of knowledge, we would prefer to retain this information.
Reviewer:
Line 785: approximately
Answer: I didn´t find any approximately in this line. Was it an orthographical correction what you meant?
Reviewer:
Please explain, why you think there were similar layers in PL and CCLR dogs. In my experience, CCLR causes severe osteoarthritis and -arthrosis whereas PL joints are mainly free from OA.
Author Response:
Thank you for your valuable comment regarding why PL and CCLR cases showed a similar number of superficial synovial cell layers. We agree that CCLR is typically associated with more advanced osteoarthritis, whereas PL joints are often described as less degenerative. However, our findings suggest that synovial remodeling is driven predominantly by chronic joint instability and persistent synovial irritation, rather than by the radiographic severity of osteoarthritis alone.
To clarify this point, we have added the following explanation to the manuscript:
New lines 795-799: “This similarity between PL and CCLR cases likely reflects the influence of chronic joint instability rather than the degree of radiographic osteoarthritis. Even though PL is often considered less degenerative, prolonged abnormal loading and repetitive soft-tissue strain can still create enough synovial irritation to elicit a hyperplastic response comparable to that seen in CCLR.”
Importantly, the present study examined histological layers of the joint capsule, not radiographic OA severity. Chronic instability—regardless of whether caused by CCLR or long-standing PL—can induce comparable microscopic alterations of the joint capsule, such as synovial proliferation, inflammation, reduced or absent subsynovial structures, and capsular remodeling. These shared biological pathways may explain why some histological layer patterns were similar in both groups. Therefore, we evaluated the clinical course and chronicity of the disease in every case to ensure that the histological findings were interpreted in the correct clinical context.
Reviewer:
Line 800: Did your clinical evaluation of the PL joint really reveal chronic synovial effusion? How old were the CCLRs that SEF values were that different to PL dogs? Please use the discussion to answer these questions, explain the context and compare to other joints / animals / humans.
Answer: We thank the reviewer for this question. We would like to clarify that we did not quantify synovial effusion in this study. The variable SEF (Surface Enlargement Factor) refers exclusively to the histological surface complexity of the synovial membrane, reflecting villus and plica formation. SEF is therefore a measure of synovial remodeling rather than joint effusion.
Clinical indicators such as swelling, pain response, and crepitus were recorded during orthopedic examination, but no imaging-based or quantitative assessment of synovial effusion was performed.
Regarding the age of the CCLR dogs, individual age values for each case were not available in the final anonymized dataset used for the histological evaluation, and therefore could not be statistically compared to PL cases. However, the epidemiology of CCLR is well established, with the condition generally occurring in middle-aged to older dogs. Combined with the chronicity categories that we did record (peracute, acute, subacute, subchronic, chronic), this supports our interpretation that many CCLR cases likely represented more chronic joint disease compared to PL cases.
Some new lines showing similarities between these results with human and equine studies have been added (New lines: 810-812 and 827-829)
Reviewer
Line 922: which ones? What is the take-home message eg. for a Chihuahua breeder?
Answer:
Thank you for this helpful comment. But I don´t find anything related to what you mentioned in the line 922, as this line is already in the conclusions. We believe that the implications for breeders should be stated more explicitly. Based on our findings, the most relevant take-home message is that structural changes of the joint capsule are already present in dogs with congenital patellar luxation, even before advanced secondary osteoarthritic changes develop. This suggests that PL is not only a mechanical malalignment but also involves early soft-tissue remodeling.
For breeders, this means that dogs showing clinical signs of PL—especially small-breed dogs such as Chihuahuas, Pomeranians, or Yorkshire Terriers—should not be used for breeding, as even low-grade or early PL is associated with measurable histological alterations of the joint capsule. Reducing the transmission of these predispositions may help lower the incidence of PL in future generations. As you asked, a new conclusion has been added, see in the new manuscript.
Reviewer:
Please use this section to conclude the results, but do not make suggestions or state the importance of further research. This is not a conclusion!!!!!
Answer: We thank the reviewer for this important remark. We agree that the Conclusions section should strictly summarize the key findings of the study and not include recommendations, implications, or statements regarding future research. Based on this comment, we have rewritten the Conclusions to concisely reflect the main histological results demonstrated in our study, without interpretation or extrapolation. The revised text now focuses exclusively on the differences observed in capsule thickness, layer organization, synovial lining, and villus formation in relation to PL, CCLR, and chronicity.
New conclusion: This study demonstrated distinct histological differences in the stifle joint capsule among dogs with patellar luxation (PL), cranial cruciate ligament rupture (CCLR), and control animals. Dogs with PL and CCLR exhibited an increased number of superficial synovial cell layers compared to controls. Chronic cases showed reduced villous for-mation as reflected by lower SEF values. Absence or marked reduction of the stratum subsynoviale was associated with PL and prolonged lameness. In PL cases, the stratum synoviale was frequently absent, whereas CCLR cases showed an increase in overall capsular thickness. These findings indicate that PL and CCLR are associated with different patterns of histomorphological remodeling of the joint capsule, and that chronicity of disease influences these structural alterations.
Reviewer:
Figure 1: different
Figure 7: green / red or better black / grey?
Answer:
Thank you for the comment. Both mistakes corrected
Reviewer 2 Report
Comments and Suggestions for Authors
Structural and histomorphological evaluation of the stifle joint capsule in canine congenital patellar luxation and cranial cruciate ligament rupture
The reviewer thanks the authors for this very interesting and thorough manuscript.
Line109 – study design accepts different grading for the patellar luxation by veterinarians. To adjust for the discrepancy the cases with multiple radings were moved into an intermediate grade – This could alter the results as different histological joint capsule features were found with different grades – this assignment of the category intermediate grade could wash out the results.
Line 126 – chronic renal diz – could that be changing the joint capsule ? –
Line 239 – proximal joint capsule on the lateral aspect of the dog with patellar luxation is expected to be different as the mechanical rubbing of the joint capsule over the femoral trochlear ridges when the patella luxates creates a thickened tissue proximally. – This mechanical stress does not occur for a CCLR case. Would it be better to only compare distal joint capsule between CCLR cases and PL cases?
Line 277 – There are 71 study dogs and only 4 control dogs. It appears the group of control dogs is small and unusual. Ideally control dogs would be similar in age size etc than the study dogs, but one patient is 0.3 years (also Line 335) old which is a very young puppy and the reviewer does not think this is a good comparison. One of the other control dogs is a dog with kidney disease. How does kidney disease affect the joint capsule integrity? The reviewer would like to see a more representative control group. Additionally, it is not clear how stifle disease was ruled out. There are plenty older patients euthanized that happen to also have chronic stifle disease.
Line 287 – The authors refer to interobserver variability in PL grading. To relate stifle joint capsule changes to patella grade there needs to be consistent reliable patella grading, which does not appear to be the case. Please explain.
Line 308 _ please provide the weight categories on the x - axis again – They are mentioned in the material and methods but it is easier for the reader and proper to place weight in kg needs to the breed category.
Line 903 In the limitation section Line 903 discusses the difficulty to obtain control patients. A suggestion of the reviewer is to perhaps collaborate with other institutions for additional controls as a representative control group is important to this study.
Author Response
Reviewer 2
We thank the reviewer for the constructive and highly valuable comments provided on our manuscript. We greatly appreciate the time and expertise invested in evaluating our work. The reviewer has highlighted several important aspects that have helped us to further clarify our methodology, refine our interpretations, and strengthen the transparency of our study. We are fully aware of the limitations inherent to our dataset, and we have carefully revised and expanded the Limitations section to ensure that readers can clearly understand these constraints and their implications. All corresponding clarifications and modifications have been incorporated into the revised manuscript.
Reviewer Comment:
“The study design accepts different grading for the patellar luxation by veterinarians. To adjust for the discrepancy, the cases with multiple readings were moved into an intermediate grade. This could alter the results as different histological joint capsule features were found with different grades. This assignment of the intermediate category could wash out the results.”
Author Response:
We thank the reviewer for this thoughtful and important comment. We agree that assigning cases with discrepant clinical grades to an intermediate category may influence the separation of grade-specific histological patterns. To address this concern, we have now added a dedicated statement to the Limitations section. This new paragraph acknowledges that the use of an intermediate category, while helpful for reducing inter-observer disagreement, may still affect the ability to detect clear histological distinctions between severity grades. We additionally note that future studies incorporating detailed assessments of structural changes in the surrounding hard tissues (e.g., femoral trochlear morphology, tibial alignment, rotational deformities) could allow for a more precise definition of patellar luxation grade and reduce grading ambiguity.
The revised text has been included in the manuscript accordingly. New lines: 908-916
Reviewer Comment:
Line 126- “Could chronic renal disease or other chronic conditions have affected the morphology of the joint capsule in your samples?
Author Response:
We thank the reviewer for raising this point. To our knowledge, chronic systemic diseases such as chronic renal disease do not directly alter the histological morphology of the joint capsule. Indirect effects—such as reduced mobility or decreased activity—may occur in chronically ill animals, but these influences are functional rather than structural and are not known to produce specific microscopic changes in the joint capsule. In contrast, chronic joint-specific conditions (e.g., osteoarthritis, immune-mediated arthritis, or long-standing joint instability) are the primary causes of capsular fibrosis, synovial hyperplasia, and collagen remodeling.
Reviewer comment:
Line 239 – proximal joint capsule on the lateral aspect of the dog with patellar luxation is expected to be different as the mechanical rubbing of the joint capsule over the femoral trochlear ridges when the patella luxates creates a thickened tissue proximally. – This mechanical stress does not occur for a CCLR case. Would it be better to only compare distal joint capsule between CCLR cases and PL cases?
Author Response:
We thank the reviewer for this important observation. It is correct that in patellar luxation the proximal–lateral joint capsule may be subject to repeated mechanical friction against the femoral trochlear ridges, which can induce localized thickening. In contrast, this specific mechanical stress does not occur in CCLR, where capsular changes are driven primarily by instability and synovitis rather than direct friction.
However, in our study the entire length of the joint capsule was collected, processed, and evaluated. Therefore, our analysis reflects the global histomorphological condition of the capsule rather than a strictly regional comparison. We agree that regional differences exist, but restricting the comparison to only the distal capsule would exclude biologically relevant information regarding proximal capsular remodeling—particularly because both PL and CCLR induce chronic instability, which affects the capsule in a more diffuse manner.
Importantly, the study still includes direct distal comparisons, and these results remain clearly interpretable within the dataset. Because our goal was to characterize the overall histomorphological remodeling patterns in PL and CCLR, we consider it appropriate to analyze the full capsular sample.
Reviewer comment:
Line 277 – There are 71 study dogs and only 4 control dogs. It appears the group of control dogs is small and unusual. Ideally control dogs would be similar in age size etc than the study dogs, but one patient is 0.3 years (also Line 335) old which is a very young puppy and the reviewer does not think this is a good comparison. One of the other control dogs is a dog with kidney disease. How does kidney disease affect the joint capsule integrity? The reviewer would like to see a more representative control group. Additionally, it is not clear how stifle disease was ruled out. There are plenty older patients euthanized that happen to also have chronic stifle disease.
Author Response:
We thank the reviewer for this valuable comment. We fully agree that the composition of the control group is important for interpreting histological differences. As requested, we have clarified the selection criteria and the rationale in the revised manuscript.
Our aim was to include dogs without any evidence of stifle disease. However, as the reviewer correctly points out, many older euthanized dogs do present with chronic stifle pathology, which significantly restricted the availability of truly unaffected stifle joint capsules. Consequently, only four dogs met the strict criteria of being orthopedically normal at the stifle.
Included sentence in the materials and methods: „For the control group, joint capsule samples were obtained from dogs euthanized for unrelated health issues, such as foreign body ingestion or age-related conditions. Orthopedic disease was ruled out through physical examination, clinical history, and owner-reported history. Owners provided written consent for the use of these samples“- New lines: 122-125
Regarding the individual cases:
The 0.3-year-old dog was included only because the stifle joint was clinically and radiographically normal. We agree that the young age is a limitation and have clearly stated this in the manuscript.
The dog with chronic kidney disease showed no orthopedic abnormalities of the stifle. To our knowledge, and based on current literature, chronic renal disease does not directly affect joint capsule histomorphology, although mild indirect effects through reduced mobility cannot be fully excluded. This is now acknowledged in the Discussion.
Importantly, the primary objective of our study was to compare histomorphological remodeling between PL and CCLR, and the control group served primarily as a reference baseline rather than a fully matched comparison cohort. We have now emphasized this point and explicitly included the small and heterogeneous control group as a limitation (Lines 918-927).
Despite these constraints, we believe the inclusion of a small reference group still provides useful contextual information, while the main conclusions of the study rely on the comparisons between PL and CCLR, where sample sizes are robust.
Reviewer comment:
Line 287 – The authors refer to interobserver variability in PL grading. To relate stifle joint capsule changes to patella grade there needs to be consistent reliable patella grading, which does not appear to be the case. Please explain.
Answer:
We thank the reviewer for this important comment. We agree that consistent and reliable patellar luxation grading is essential when attempting to relate stifle joint capsule changes to PL severity. Interobserver variability in PL grading is a well-recognized limitation in both clinical and research settings. In our dataset, a small number of dogs received discrepant grades from different clinicians. To handle this variability conservatively, we assigned such cases to an intermediate category rather than forcing them into a potentially inaccurate grade. This approach allowed us to include these cases while avoiding misclassification bias.
We acknowledge that this variability may reduce the precision of grade-specific associations. We have clarified this point in the manuscript in the discussion/limitation. As noted earlier, future studies incorporating objective assessments of underlying skeletal morphology—such as femoral trochlear shape, tibial alignment, and rotational deformities—could help refine and standardize PL grading, thereby reducing interobserver differences and improving the reliability of such correlations (Lines 908-916)
Reviewer: Line 308 _ please provide the weight categories on the x - axis again – They are mentioned in the material and methods but it is easier for the reader and proper to place weight in kg needs to the breed category.
Answer: Thanks a lot for this comment. We improved the figure and included the weight in every mentioned breed group. New Figure 6
Reviewer comment:
Line 903 discusses the difficulty to obtain control patients. A suggestion of the reviewer is to perhaps collaborate with other institutions for additional controls as a representative control group is important to this study.
Answer:
We appreciate the reviewer’s suggestion regarding collaboration with other institutions to increase the number of control samples. As noted, our control group is indeed very small due to the considerable difficulty in obtaining truly unaffected stifle joint capsules. Control samples were limited to dogs with distal femoral fractures or those euthanized for unrelated health conditions, and all were included only after owner consent. The inclusion of completely healthy dogs is ethically challenging, as most owners are understandably reluctant to allow tissue collection without medical necessity or after euthanasia.
We have now clarified this point in the Limitations section. While we fully agree that a larger and more representative control group would strengthen future studies, such sampling is constrained by ethical and practical limitations. Collaborative work with additional institutions may help to overcome these challenges, and future research should aim to include more control cases to establish a more robust baseline for comparison. These limitations highlight the need for further investigation to expand upon the findings of the present study (New Lines 918-927)
Round 2
Reviewer 1 Report
Comments and Suggestions for Authors
Thank you for the revisions you provided. This is really an interesting topic and study! I have only two further comments:
line 49: The three references you provided do not focuse on the hereditary CCLR but only on patellar luxation. Please provide literature to CCLR.
line 785 (original manuscript), line 787 (revised manuscript): aproximately
Author Response
Dear Reviewer,
I thank you very much the comments and remarks that improved our manuscript. Here the answers to your comments.
Thank you for the revisions you provided. This is really an interesting topic and study! I have only two further comments:
line 49: The three references you provided do not focuse on the hereditary CCLR but only on patellar luxation. Please provide literature to CCLR.
Answer: As reference I added now citation number 4- Kowaleski
line 785 (original manuscript), line 787 (revised manuscript): aproximately
Answer: Approximately has now been corrected.
Thank you very much!
Reviewer 2 Report
Comments and Suggestions for Authors
The reviewer thanks the authors for the meaningful changes made to the manuscript.
There are only a couple comments remaining.
Despite these constraints, we believe the inclusion of a small reference group still provides useful contextual information, while the main conclusions of the study rely on the comparisons between PL and CCLR, where sample sizes are robust. –
A reference group is important and is not in question. Only the reference group in this manuscript is not well matched with the clinical groups. Age matched clinical healthy dogs with similar body weight to the mean of the individual groups would have been best. The reviewer understands the limitation to the control group case selection. Unnecessary procedures on healthy patients are not suggested. Please strike – “without medical indication or” from sentence below.
Line 919 - Obtaining completely healthy stifle joint capsules is ethically challenging, as owners are generally reluctant to allow tissue collection without medical indication or after euthanasia
The authors write:
Line 122-125 For the control group, joint capsule samples were obtained from dogs euthanized for unrelated health issues, such as foreign body ingestion or age-related conditions. Orthopedic disease was ruled out through physical examination, clinical history, and owner-reported history. Owners provided written consent for the use of these samples.
A suggestion is to radiograph the stifle in addition to a physical examination to document a normal stifle which could be done after the patient is diseased. This is just a comment and no further changes to the manuscript are required.
Author Response
Dear Reviewer 2,
I thank you very much for your remarks and comments which have improved the quality of our manuscript.
Here the two answers to your comments.
Reviewer 2:
A reference group is important and is not in question. Only the reference group in this manuscript is not well matched with the clinical groups. Age matched clinical healthy dogs with similar body weight to the mean of the individual groups would have been best. The reviewer understands the limitation to the control group case selection. Unnecessary procedures on healthy patients are not suggested. Please strike – “without medical indication or” from sentence below.
Line 919 - Obtaining completely healthy stifle joint capsules is ethically challenging, as owners are generally reluctant to allow tissue collection without medical indication or after euthanasia
Answer: Line 919 has been corrected as proposed by reviewer 2, striking "Without medical indication or"
Comment:
Line 122-125 For the control group, joint capsule samples were obtained from dogs euthanized for unrelated health issues, such as foreign body ingestion or age-related conditions. Orthopedic disease was ruled out through physical examination, clinical history, and owner-reported history. Owners provided written consent for the use of these samples.
A suggestion is to radiograph the stifle in addition to a physical examination to document a normal stifle which could be done after the patient is diseased. This is just a comment and no further changes to the manuscript are required.
Answer: I agree that an x-ray of the stifle joint would have been very valuable in this case and is one of the limitations in general of this study. A radiological study of all these stifle joints, with PL, CCLR or controls would have been very useful to correlate it with osteoarthrosis.
Thank you for your remarks and comments!
Best regards,